# PROMPTING FAIRNESS: INTEGRATING CAUSALITY TO DEBIAS LARGE LANGUAGE MODELS

**Jingling Li**[*1], **Zeyu Tang**[*2], **Xiaoyu Liu**[3], **Peter Spirtes**[2], **Kun Zhang**[2,4], **Liu Leqi**[5], **Yang Liu**[6]

[1]Google DeepMind
[2]Department of Philosophy, Carnegie Mellon University
[3]Department of Computer Science, University of Maryland, College Park
[4]Machine Learning Department, Mohamed bin Zayed University of Artificial Intelligence
[5]University of Texas at Austin
[6]Computer Science and Engineering Department, University of California, Santa Cruz

## ABSTRACT

Large language models (LLMs), despite their remarkable capabilities, are susceptible to generating biased and discriminatory responses. As LLMs increasingly influence high-stakes decision-making (e.g., hiring and healthcare), mitigating these biases becomes critical. In this work, we propose a causality-guided debiasing framework to tackle social biases, aiming to reduce the objectionable dependence between LLMs' decisions and the social information in the input. Our framework introduces a novel perspective to identify how social information can affect an LLM's decision through different causal pathways. Leveraging these causal insights, we outline principled prompting strategies that regulate these pathways through selection mechanisms. This framework not only unifies existing prompting-based debiasing techniques, but also opens up new directions for reducing bias by encouraging the model to prioritize fact-based reasoning over reliance on biased social cues. We validate our framework through extensive experiments on real-world datasets across multiple domains, demonstrating its effectiveness in debiasing LLM decisions, even with only black-box access to the model.

## 1 INTRODUCTION

Large language models (LLMs) trained on massive text corpora have been found to exhibit concerning levels of social biases (Sheng et al., 2019; Gonen & Goldberg, 2019; Schick et al., 2021; Bender et al., 2021; Dodge et al., 2021). The unchecked biases can potentially perpetuate and amplify societal inequities, leading to unfair or even unethical outcomes. This issue is particularly significant as LLMs become more capable and start to serve as foundational components in decision-making systems across various sectors such as healthcare and education. Many debiasing approaches have been proposed to tackle this issue, for instance, direct fine-tuning of model parameters (Kaneko & Bollegala, 2021; Garimella et al., 2021; Lauscher et al., 2021; Guo et al., 2022), modifying the decoding steps (Schick et al., 2021), and prompting-based techniques (Si et al., 2022; Tamkin et al., 2023; Oba et al., 2023; Ganguli et al., 2023). For various reasons such as security and business interests, many of the most capable LLMs are closed-source models, for instance, GPT-4 (OpenAI, 2023), Gemini (Anil et al., 2023), Claude (Anthropic, 2024), where the general public do not have access to models' internal structures or parameters. Therefore, prompting-based techniques largely become the only viable option to mitigate bias when using closed-source LLMs.

In this work, we focus on prompting techniques that mitigate bias in LLMs' decisions. We observe that unbiased decision-making in an LLM essentially hinges on the *selection* and utilization of appropriate dependence patterns within its internal representations or knowledge. Let us consider a simple coreference resolution task on gender pronouns in a given sentence. A model may output

---

[*]Equal contribution. Correspondence to: Jingling Li `<jinglingli1024@gmail.com>` and Zeyu Tang `<zeyutang@cmu.edu>`. The work was done while JL and YL were working at ByteDance Research.

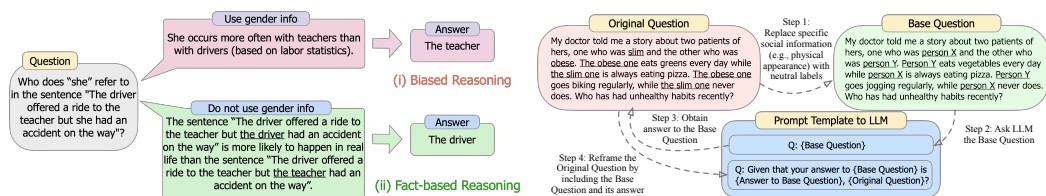

(a) Different reasoning to a coreference question  (b) A systematic way to promote fact-based reasoning

Figure 1: Panel (a): A biased answer may be due to the use of a gender shortcut, while a fact-based answer is made by considering proper world knowledge given the circumstances. Panel (b): we describe how to systematically generate prompts that encourage fact-based reasoning. We would like to note that using social category information does not necessarily indicate that the reasoning is biased: sometimes, certain social category information *should* be considered, e.g., gender in medical treatments. We call such dependence neutral dependence, while in this work, we focus on objectionable/problematic ones (Tang et al., 2024).

a biased answer via a gender shortcut potentially learned from the unbalanced training data. For example, in Figure 1(a), the model may associate the pronoun "she" with "teacher" rather than "driver" due to social biases reflected in the training data–the gender distributions are disproportional for different occupations. We denote such reasoning process as *biased reasoning*, where the model *selects* and utilizes inappropriate dependence patterns involving specific social information (gender in this example) that should actually be irrelevant to solving the coreference task.

Most existing prompting-based debiasing methods focus on *discouraging biased reasoning*, for instance, using explicit instructions to avoid the usage of biased association when making the decision. While these approaches help to a certain extent, another overlooked debiasing strategy is to *encourage fact-based reasoning* by only *selecting* proper dependence patterns related to the nature of the question. For example, in Figure 1(a), we ask the model to first compare the likelihood of two situations occurring in real life and draw on the more likely situation to infer the coreference resolution. In Figure 1(b), we also provide a systematic approach to encourage fact-based reasoning.

We formalize the above intuitions into a causality-guided debiasing framework, offering a novel perspective on how social information influences an LLM's decision-making through different causal pathways. Within this proposed framework, we demonstrate how *selection mechanisms* can introduce bias into the training data while also help mitigate bias in LLMs' decisions. Specifically, we present principled prompting-based strategies that regulate different causal pathways via employing selection mechanisms, and these strategies can be applied to (1) discourage biased reasoning, and (2) encourage fact-based reasoning. Current prompting-based debiasing methods can be viewed as instantiating one or both of these goals.

We conduct extensive empirical studies on real-world datasets across multiple domains, and our proposed strategies significantly outperform existing prompting-based debiasing methods. The strong empirical results clearly demonstrate the effectiveness of our causality-guided debiasing framework, even when we only have black-box accesses. Our contributions can be summarized as follows:

- We develop a causality-guided debiasing framework that provides a novel perspective on how social information can influence LLMs' decisions through different causal pathways.
- Using the proposed framework, we present principled prompting-based debiasing strategies that utilize *selection mechanisms* to regulate the influence of bias. These strategies aim to discourage biased reasoning and encourage fact-based reasoning, offering a unified view of existing prompting-based methods and addressing the gap in promoting fact-based reasoning.
- We conduct extensive empirical studies and demonstrate the effectiveness of our framework in debiasing LLMs' decisions across various social categories, highlighting both its theoretical foundations and practical significance in debiasing LLMs' decisions.

## 2    PRELIMINARIES

In this section, we introduce our problem setting, defining what we mean by an unbiased decision for large language models (Section 2.1). Then we provide a brief introduction to causal modeling and causal reasoning (Section 2.2). We also provide a motivating example to illustrate how the selection mechanism can reshape dependence patterns among different variables (Section 2.3).

## 2.1 PROBLEM SETTINGS

Our paper focuses on steering LLMs to make unbiased *decisions*, which is a more specific task than de-biasing LLMs' outputs or responses. Let $Y$ denote the decision an LLM can make given a context, and let $\mathcal{A}$ denote the set of social dimensions we want to consider in mitigating social biases (e.g., age, race, gender, etc.). Let $A \in \mathcal{A}$ denote a specific social dimension (e.g., race) in the context that should not influence the decision-making process. Then, an unbiased decision means that $Y \perp\!\!\!\perp A$. This definition resembles dependence-based fairness definitions (e.g., *statistical parity*) for classification/regression models (Calders et al., 2009; Kamiran & Calders, 2009; Lum & Johndrow, 2016; Jiang et al., 2022).

## 2.2 A BRIEF INTRODUCTION TO CAUSALITY

For two random variables $X$ and $Y$, $X$ is a direct cause of $Y$ if there is a change in the distribution of $Y$ when we apply an intervention on $X$ while holding all other variables fixed (Spirtes et al., 1993; Pearl, 2009). We can represent causal relations among variables with a directed acyclic graph (DAG), where nodes represent variables, and edges represent direct causal relations between variables. We denote the direct causal relation between the ordered pair $(X, Y)$ by a directed edge $X \to Y$. In the context of language processing, the definition of a variable representing text or tokens is relatively abstract compared to the statistical notion of a random variable in tabular data. Within the scope of this work, we use the terms "variable" and "node" interchangeably when the context permits clear understandings.

## 2.3 REMARKS ON SELECTION MECHANISMS: AN ILLUSTRATIVE EXAMPLE

Ideally, we would like samples to be drawn uniformly from the underlying population of interest. However, in practice, the probability of including certain data points in the training corpus often depends on their characteristics where selection mechanisms are commonly involved.[1]

Consider an example in the context of medical care: as illustrated in Figure 2, the observed variables $(X_1, X_2)$ denote diseases and $(Y_1, Y_2)$ represent corresponding symptoms. There can exist potential binary selection variables $S_i$'s ($i \in \{1, 2, \ldots, 5\}$), where $S_i = 1$ indicates the occurrence of that selection. $X_1$ ($X_2$) is the direct and only cause of $Y_1$ ($Y_2$), and the two diseases are uncorrelated in the general population, i.e., when none of the $S_i$'s exists. Solid edges in the figure denote causal relations among observed variables, while dashed edges represent the ones pertaining to selection mechanisms.

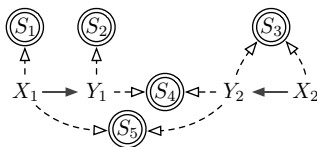

Figure 2: An illustrative example of selection mechanisms.

For instance, $S_2$ is an example of outcome-dependent selection (Zhang et al., 2016), which can be selecting individuals with symptom $Y_1$ from the general population. The conditional probability $P(Y_1 \mid X_1, S_2 = 1)$ in the selected data typically differs from its counterpart $P(Y_1 \mid X_1)$ in the general population (Zhang et al., 2016). As another example, $S_4$ demonstrates the selection solely based on the hospital in-patient data, a setting of the well-known Berkson's Paradox (Berkson, 1946), where two unrelated diseases $(X_1, X_2)$ appear to be correlated in the hospital data, simply because the sample includes only admitted patients who have at least one symptoms in $(Y_1, Y_2)$.

Selection can be solely based on the cause (e.g., $S_1$), solely based on the effect (e.g., $S_2$), or both (e.g., $S_3$), as presented in Figure 2. Meanwhile, the selection can be on variables that are not causally related in the general population (e.g., $S_4$ and $S_5$). Because selection mechanisms can reshape dependence patterns among variables, they can also introduce (conditional) independence relations that are not entailed by Markov conditions on the graph.[2] For instance, while $X_2$ is a direct cause of $Y_2$, the presence of $S_3$ can make $X_2$ and $Y_2$ appear independent after selection. Selection through $S_3$ based on particular value combinations of $(X_2, Y_2)$ at specific proportions can instantiate this phenomenon. We will see in Section 3 that such property of selection also applies to natural language processing (NLP) contexts. The ability to reshape dependence patterns via selection mechanisms can be leveraged to debias LLM decisions. By designing prompt strategies that enforce the independence between the social category information and LLM decisions ($Y \perp\!\!\!\perp A$), we can mitigate biases effectively.

---

[1]A thorough literature review on selection mechanisms can be found in Appendix A.1.

[2]This phenomenon is an instance of violation of *Faithfulness* assumption (Spirtes et al., 1993).

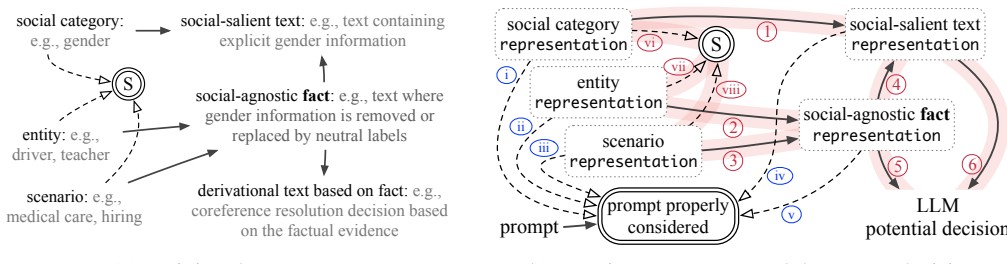

(a) Training data corpus

(b) How input prompts modulate LLM decisions

Figure 3: Causal graphs for different data generating processes. We use double-stroke contours to indicate selection variables, solid edges to represent causal relations among observed variables or internal representations, and dashed edges for those pertaining to selection mechanisms. Panel (a) presents the underlying data generating process of the training data corpus. Panel (b) presents a causal perspective on how LLM's decisions are related to internal representations and modulated by external prompts. We highlight in light coral to denote pathways along which social category information can influence the LLM's decision.

# 3    CAUSALITY-GUIDED DEBIASING FRAMEWORK

This section presents our causality-guided debiasing framework. Specifically, we provide a causal modeling on the data generating process of LLMs' training corpora in Section 3.1, highlighting how bias can be introduced into the training data by selection mechanisms. In Section 3.2, we demonstrate how social category information can impact LLMs' decisions via different causal pathways. Section 3.3 further reveals how selection mechanisms can regulate the information flow over these pathways by reshaping dependence patterns. The regulations over different causal pathways leads to prompting strategies with distinct objectives aimed at effectively mitigating biases in LLMs. Moreover, we prove that when these objectives are satisfied, we can completely remove the influence of social category information from the LLMs' decisions, which allows the LLMs' decision to be conditionally independent of the social information, given the presence of appropriate selection mechanisms.

## 3.1    UNDERLYING DATA GENERATING PROCESS OF LLMS' TRAINING CORPORA

The training data of LLMs often reflects historical discrimination or stereotypes, and selection mechanisms are commonly involved. For instance, gender stereotype in occupations arises not due to the existence of a direct causal relation or a common cause between gender and occupation, but rather from an underlying selection mechanism: in the training data, among all possible combinations between gender (e.g., male and female) and occupation (e.g., CEO and secretary), there is a tendency to associate CEO more often with male, and secretary with female due to historical or societal biases. This phenomenon may occur when the training data is a subset *selected* from an imaginary data corpora that is ideally diverse and comprehensive, where there is no historical discrimination or stereotype when people generate text content. Therefore, the selection mechanism between gender and occupations introduces bias into the training data, which was fed to LLMs during the pretraining stage.

More generally, in Figure 3(a), we provide a causal graph to denote the underlying data generating process of training corpora (e.g., text scraped from the internet). The causal graph contains additional variables of interest besides information related to a *social category* (e.g., age. gender, and race): "*scenario*" represents the practical situation or context where a decision is needed (e.g., medical care, hiring); "*entity*" denotes the participants or stakeholders involved in the scenario (e.g., a patient in the medical care scenario, or a secretary in the hiring scenario); and the *selection variable S* explicitly models the associations of social categories with particular entities and scenarios, which are recognized as a major type of stereotypes in NLP (Sweeney, 2013; Bolukbasi et al., 2016; Zhao et al., 2018; Tamkin et al., 2023). There are also different types of texts in Figure 3(a): "*social-agnostic fact*" denotes the text that does not explicitly contain information related to a social category of interest (e.g., base question in Figure 1(b)); "*social-salient text*" denotes the text where information related to the social category appears explicitly (e.g., original question in Figure 1(b)); and "*derivational text based on fact*" denotes the text or decision derived from factual evidence (e.g., inference according to fact-based reasoning).

## 3.2 How Social Category Information Can Impact LLM's Decisions

The causal modeling of the underlying data generating process in Section 3.1 characterizes how bias could be introduced into LLMs' training data. In this subsection, we explore how LLM outputs can be potentially shaped and modulated by different prompt designs through selection mechanisms, as presented in Figure 3(b). We highlight certain edges in light coral (accompanied by annotations marked with circled red numerals) to denote unregulated information flow from social category representations to LLM's output, which potentially leads to biases in the output decisions. We use dotted contours to distinguish LLMs' internal representations from the actual external information in the data corpora, e.g., the contrast between "*social category*" in Figure 3(a) and "*social category representation*" in Figure 3(b). Notice that the internal nodes are not directly observable or accessible. For the selection mechanisms in Figure 3(b), we use circled red Roman numerals to represent ones corresponding to historical discrimination (Section 3.1), and we use circled blue Roman numerals to denote the ones that can be specified by the external input prompt.

In Figure 3(b), a "*prompt*" serves as an external input to LLM rather than a direct cause of changes in internal representations, because the internal knowledge and representations exist *a priori*, and whether a specific prompt is provided does not alter their existence. The prompt also does not act as an indicator for causal interventions, since internal nodes are not directly observable or accessible, and one cannot set them to certain values via hard interventions (Spirtes et al., 1993; Pearl, 2009; Peters et al., 2017; Hernán & Robins, 2020), or change their functional behavior via soft interventions (Eberhardt & Scheines, 2007; Huang et al., 2020; Correa & Bareinboim, 2020). However, the input prompt directly changes the selection variable "*prompt properly considered*" (PPC), a concept which we further examine in Section 3.3. If the LLM model is well-trained and well-aligned, we can expect that the model will always condition on PPC when producing a "*LLM's potential decision*."

## 3.3 Causality-Guided Debiasing Strategies

As illustrated in Figure 3(b), social category information can influence the LLM's potential decisions via two important nodes: the "social-agnostic fact representation" node and the "social-salient text representation" node. Our core idea is to formulate objectives that regulate the information flow through these two nodes. By designing prompts that satisfy these objectives, we can effectively debias LLM decisions through selection mechanisms.

Regulating the information flow through the "social-agnostic fact representation" node leads to prompt-design strategies aimed at encouraging fact-based reasoning (Strategy I and Strategy II), and regulating the flow through the "social-salient text representation" node results in strategies that discourage biased reasoning (Strategy III).

**Strategy I (*Nudge Towards Social-Agnostic Fact*).** Strategy I steers LLMs towards relying more on the social-agnostic fact when making decisions. The **objective** is to utilize the selection mechanism over internal representations of social category (connected via edge ⓘ) and social-agnostic fact (connected via edge ⓥ), aiming to make them conditionally independent in the presence of PPC and the existing selection $S$:

$$\underset{\text{representation}}{\text{social category}} \perp\!\!\!\perp \underset{\text{fact representation}}{\text{social-agnostic}} \mid S = 1, \text{PPC} = 1. \tag{1}$$

Strategy I focuses on social-agnostic fact, which is a direct parent of LLM potential decision. The objective is to encourage LLMs to utilize social-agnostic fact, while regulating its dependence on social category through the selection mechanism involving these two (denoted by edges ⓘ and ⓥ). The ideal outcome is that the information flow through social-agnostic fact to downstream nodes (along edges ④ and ⑤) is regulated (see Figure 4(b)). *An example prompt* employing Strategy I can be: "Considering the fact that the sentence 'the driver offered a ride to the teacher but the driver had an accident on the way' is more likely to happen in real life than the other sentence 'the driver offered a ride to the teacher but the teacher had an accident on the way,' who does 'she' refer to in the sentence 'the driver offered a ride to the teacher but she had an accident on the way'?"

**Strategy II (*Alternative Fact Counteracts Existing Selection Bias*).** Strategy II directly counteracts existing bias instantiated by the existing selection $S$ (through edges {ⓥⓘ, ⓥⓘⓘ, ⓥⓘⓘⓘ}). The **objective** is to use selection mechanisms over internal representations of social categories, entities, scenarios (through edges {ⓘ, ⓘⓘ, ⓘⓘⓘ}) to enforce conditional independence between entity and social category,

and between scenario and social category, in the presence of PPC and the selection $S$:

$$\begin{aligned} \text{social category representation} &\perp\!\!\!\perp \text{entity representation} \mid S=1, \text{PPC}=1, \\ \text{social category representation} &\perp\!\!\!\perp \text{scenario representation} \mid S=1, \text{PPC}=1. \end{aligned} \tag{2}$$

Strategy II aims to counteract the existing selection bias from $S$ by introducing corresponding selection mechanisms that simultaneously operate over internal representations for social category (by PPC via edge ⓘ and by $S$ via edge ⓥⓘ), entity (by PPC via edge ⓘⓘ and by $S$ via edge ⓥⓘⓘ), and scenario (by PPC via edge ⓘⓘⓘ and by $S$ via edge ⓥⓘⓘⓘ). The ideal outcome is that the downstream node of entity and scenario representations, i.e., social-agnostic fact, does not inherit dependence on the social category from its direct parents (along edges ② and ③; see Figure 4(c)). *An example prompt* employing Strategy II can be: "Assume male and female are equally represented in drivers and in teachers. Who does 'she' refer to in the sentence 'the driver offered a ride to the teacher but she had an accident on the way'?" despite that the actual representation may be highly uneven in the historical data used in pretraining.

**Strategy III** (*Nudge Away from Social-Salient Text*). Strategy III steers LLMs away from utilizing social-salient information when making decisions. The **objective** is to utilize the selection mechanism over internal representations of social categories (via edge ⓘ) and social-salient text (via edge ⓘⓥ) to make them conditionally independent in the presence of PPC and the existing selection $S$:

$$\text{social category representation} \perp\!\!\!\perp \text{social-salient text representation} \mid S=1, \text{PPC}=1. \tag{3}$$

Strategy III focuses on social-salient text, which is a direct parent of LLM potential decision. Since social-salient text inherently contains information about social category, Strategy III essentially discourage the use of social-salient text when forming decisions. This is achieved by regulating the information flow between social category and LLM potential decision through edges ① and ⑥ (see Figure 4(d)). *An example prompt* employing Strategy III can be: "Do not answer the question using gender information. Who does 'she' refer to in the sentence 'the driver offered a ride to the teacher but she had an accident on the way'?"

**Remarks on the Three Strategies.** While each of the three strategies is individually effective to some extent, none is perfect in isolation.[3] For instance, although Strategy I steers LLMs towards using social-agnostic facts, it does *not* explicitly prevent LLMs from using social-salient representations in making decisions. Consequently, the social category can still influence the output decision through an unregulated path involving edges ① and ⑥. Similarly, while Strategy II aims to regulate information related to social category representation that flows from representations of entity and scenario to downstream along edges {②, ③}, there is *no* explicit constraint involving the information flow along edges ① and ⑥, or along edges ④ and ⑤, which leaves space for bias to sneak in.

In contrast to Strategy I, which encourages LLMs to focus on social-agnostic facts, Strategy III prevents LLMs from relying on social-salient text during reasoning. Although edge ① is explicitly regulated by the selection mechanism denoted by edges ⓘ and ⓘⓥ, the information flow along edge ⑤ can still result in the association between social category and the output decision. This is because Strategy III does *not* involve objectives to change the dependence patterns between social category and social-agnostic fact. As seen from the objective specified in Strategy II, the absence of explicit social category information in a social-agnostic fact does not guarantee the absence of dependence between the two, since direct parents (entity and scenario representations) of social-agnostic fact may still be dependent with the social category if without explicit constraints. Debiasing is better realized when these strategies are combined to address social bias in LLMs more comprehensively:

**Theorem 3.1** (**Comprehensive Debiasing When Combining All Three Strategies**). *If conditions and constraints specified in Equations (1), (2), and (3) are simultaneously satisfied, the LLM's decision $Y$ is independent from the social category $A$ in the presence of PPC and existing selection $S$:*

$$Y \perp\!\!\!\perp A \mid S=1, \text{PPC}=1. \tag{4}$$

*Remark* 3.2. The PPC here is the selection node representing multiple selection mechanisms simultaneously, with the objectives in all three prompting strategies fully achieved.[4]

---

[3]We provide additional illustrations in Appendix B.

[4]We implicitly utilize the mild assumption that a well-trained and well-aligned LLM captures the dependence pattern in the training data and that such a pattern is internalized and utilized during reasoning. Though it may

*Proof.* There are two direct parents of LLM's decision $Y$. To enforce the independence between $Y$ and the social category $A$, a sufficient condition is that both direct parents of $Y$ are conditionally independent of $A$, in the presence of proper PPC and the existing selection $S$. For the social-agnostic fact, considering its direct parents–representations for entity and scenario–the conditional independence can be indirectly enforced through Equation (2), and further directly guaranteed by Equation (1). For social-salient text, the conditional independence can be enforced by Equation (3). Therefore, Equations (1), (2), and (3) altogether removes the (objectionable) dependence between social category information and LLM decisions, as described in Equation (4). □

Theorem 3.1 indicates that when jointly employing Strategies I, II, and III, we can successfully achieve the debiasing of LLM decisions. From a purely technical standpoint, Equation (1) and Equation (2) might seem redundant, as both of them aim to enforce the conditional independence between social category and social-agnostic fact given PPC and $S$. However, because of the definition of a variable in the language context is relatively abstract, it is favorable to have empirical strategies that can debias LLM decisions from different perspectives. In Section 4, we empirically demonstrate the advantage of our strategies, when utilized individually and in combination.

## 4 RESULTS AND ANALYSES

In this section, we measure how (1) encouraging fact-based reasoning (Strategies I and II), and (2) discouraging biased reasoning (Strategy III), effectively steer LLMs towards making unbiased decision across various social dimensions. We conduct extensive experiments on three widely utilized benchmarks that evaluate language models' decision bias: WinoBias by Zhao et al. (2018) (Section 4.1), the Bias Benchmark for QA (BBQ) by Parrish et al. (2021) (Section 4.2), and Discrim-Eval by Tamkin et al. (2023) (Appendix D.2). On all three benchmarks, our framework leads to strategies effectively reduce LLMs' decision biases across different social categories.

### 4.1 GENDER BIAS: WINOBIAS

The WinoBias dataset (Zhao et al., 2018) evaluates how likely models assign stereotypical gender pronouns to occupations in coreference resolution tasks. The sentences in WinoBias are designed to be structurally parallel but differ in gender pronouns. It contains two sets of sentences: *pro sentences* with the pro-stereotypical gender pronouns (e.g., nurses as she, engineers as he), and *anti sentences* with anti-stereotypical gender pronouns (e.g., nurses as he, engineers as she). The dataset also has two types of tasks with different levels of difficulties: coreference decisions in Type I task are challenging and mandate world knowledge to reason about given circumstances, whereas Type II task can be resolved using only syntactic information.

#### 4.1.1 EXPERIMENTAL SETTINGS (WINOBIAS)

For each sentence in WinoBias, we define the `original question` to be "Who does {gender pronoun} refer to in the sentence '{original sentence}'?", and we measure the performance of four large language models: GPT-3, GPT-3.5, GPT-4, and Claude 2 across the above two types of coreference tasks (Type I and Type II).

**Baseline Approaches** There are a few existing works employing prompting techniques to mitigate the bias in LLMs. We consider these three baselines:

- `Default`: asks the `original question` directly without any additional prompts.
- `ICL with contrastive examples`: provides pairs of contrastive examples followed by the `original question`. The pair of questions are contrastive as the pronouns are different but the answer remains unchanged. In WinoBias, we use the same 16 ICL examples as in Si et al. (2022). Oba et al. (2023) also used similar counterfactual preambles to suppress bias in LLMs.
- `Zero-shot COT` (Kojima et al., 2022): asks the model to also "think step by step" after the `original question`. Shaikh et al. (2022); Ganguli et al. (2023) designed similar COT approaches to mitigate bias in LLMs.

be hard to realize full achievements in practice, we show in our experiments that even if we cannot fully satisfy any of the three objectives, we can still design prompts following the proposed strategies to effectively debias LLMs' decisions.

Table 1: **Performance comparison of various debiasing methods on WinoBias.** We show that combining *encouraging fact-based reasoning* and *discouraging biased reasoning* together (i.e., DDP) significantly alleviates the gender bias for both coreference tasks. "*Pro*" in the table stands for coreference with pro-stereotypical pronouns, and "*Anti*" stands for coreference with anti-stereotypical pronouns. A smaller gap indicates less bias.

| Accuracy (%) | GPT-3 | | | GPT-3.5 | | | Claude 2 | | | GPT-4 | | |
|---|---|---|---|---|---|---|---|---|---|---|---|---|
| | Anti | Pro | Gap↓ | Anti | Pro | Gap↓ | Anti | Pro | Gap↓ | Anti | Pro | Gap↓ |
| Type I | | | | | | | | | | | | |
| Default | 43.01 | 79.24 | 36.23 | 62.96 | 94.03 | 31.07 | 67.57 | 92.13 | 24.56 | 82.50 | 97.96 | 15.47 |
| COT (zero shot) | 41.79 | 75.85 | 34.06 | 62.14 | 90.64 | 28.49 | 70.56 | 91.59 | 21.03 | 84.40 | 95.66 | 11.26 |
| ICL (16-shot) | 46.81 | 94.57 | 47.76 | 45.18 | 92.81 | 47.63 | 73.68 | 92.27 | 18.59 | 88.87 | 98.10 | 9.23 |
| DDP (ours) | 73.27 | 73.95 | **0.68** | 72.73 | 84.67 | **11.94** | 74.08 | 75.17 | **1.09** | 94.57 | 96.74 | **2.17** |
| Type II | | | | | | | | | | | | |
| Default | 85.01 | 97.97 | 12.96 | 94.28 | 98.98 | 4.70 | 93.77 | 97.46 | 3.68 | 97.59 | 99.49 | 1.91 |
| COT (zero shot) | 69.12 | 86.79 | 17.66 | 93.90 | 98.35 | 4.45 | 95.30 | 99.62 | 4.32 | 97.97 | 99.62 | 1.65 |
| ICL (16-shot) | 94.41 | 99.62 | 5.21 | 95.43 | 98.98 | 3.56 | 94.03 | 97.84 | 3.81 | 98.35 | 99.87 | 1.52 |
| DDP (ours) | 83.23 | 84.24 | **1.02** | 92.12 | 94.41 | **2.29** | 82.34 | 85.26 | **2.92** | 99.62 | 99.75 | **0.13** |

**Evaluation Metrics**   We measure the accuracy of LLMs on *pro* and *anti sentences*, and the gap between the two indicates the level of gender bias exhibited by the models (smaller gaps are better).

**Dual Directional Prompting**   We propose Dual Directional Prompting (DDP) as one way of combining different prompting strategies. Following Figure 1, we construct the base question by first creating two gender-agnostic sentences via replacing the gender pronoun with the two occupations appeared in the original sentence, and then asking the model which sentence is more likely to happen in real life. This base question does not contain any gender-related information. We then prepend the model's answer (to the base question) before asking the original question. This allows us to distill LLM's non-gender-related world knowledge and nudge it to explicitly reason with this fact during coreference resolution (Strategy I). On top of Strategy I, we also add the prompt that both occupations are equally likely to be male or female to in order to counteract existing selection bias (Strategy II). Note that the generation of base questions can be done by regular expression or one additional LLM query. For efficiency, we can also query a smaller LLM to generate the base question given the original one and the related social category.

One caveat of utilizing LLM's world knowledge is that the performance of Dual Directional Prompting will increase as the capabilities of the LLM grow, since better world knowledge will further help LLM answer both factual and original questions.

### 4.1.2   OUR RESULTS (WINOBIAS)

We compare DDP with the three baseline approaches on both coreference tasks and summarize our results in Table 1. Because of the existence of historical biases in LLMs' training data, we expect all LLMs to score higher accuracies on sentences containing pro-stereotypical pronouns (*pro sentences*) compared to these with anti-stereotypical pronouns (*anti sentences*).

**Coreference Resolution Requiring World Knowledge**   For Type I tasks, world knowledge is required to perform coreference resolution. The Default prompting shows significant biases, with large gaps in accuracy between *pro sentences* and *anti sentences* for all models. The method Zero-shot COT marginally reduces the bias, as seen in the smaller gaps, but its performance is lower on both pro and anti scenarios for GPT-3 and GPT-3.5 when compared with Default, and it only marginally improves the general performance when applied to more capable ones (Claude 2 and GPT-4). For ICL with contrastive examples, although the gap becomes larger for less capable models (GPT-3 and GPT-3.5), it can further reduce biases with more capable LLMs (Claude 2 and GPT-4), especially for GPT-4 (with a gap of 9.23%).

Remarkably, Dual Directional Prompting, which employs multiple strategies to encourages fact-based reasoning substantially decreases the bias across all LLMs. They achieved minimal gaps, with GPT-4 exhibiting a mere 2.17% gap and 94.57% accuracy on *anti sentences*, demonstrate significantly greater effectiveness compared to other approaches.

**Coreference Resolution with Only Syntactic Information**   For Type II tasks, coreference can be resolved with only syntactic cues; thus, all models generally show higher accuracies with smaller gaps. However, since there exists a shortcut in the Type II task (the correct answer is always the second

Table 2: **Error Analysis on WinoBias Type I coreference task.** We divide the models' responses into 4 categories to better understand the cause of their errors and success. **TT** denotes the (%) of examples where LLM answers both the base question and the original question correctly, **FF** denotes the (%) of examples where both questions are answered incorrectly, indicating the coreference errors caused by the model's **world knowledge**. **TF** denotes the coreference errors caused by **gender bias** as only base questions are correctly answered, and **FT** indicates coreference success that may be due to **gender shortcut** since the base questions get wrong but original questions are correctly answered. More details about the settings can be found in Appendix C.1.

| Accuracy (%) | TT | | TF | | FT | | FF | |
|---|---|---|---|---|---|---|---|---|
| | Anti | Pro | Anti | Pro | Anti | Pro | Anti | Pro |
| GPT-3.5 | | | | | | | | |
| DDP | 70.15 | 78.70 | 10.58 | 2.04 | 2.58 | 5.97 | 16.55 | 13.16 |
| Fact Only | 61.87 | 79.24 | 18.86 | 1.63 | 2.31 | 7.46 | 16.96 | 11.67 |
| Counteract Only | 42.33 | 66.49 | 32.02 | 7.60 | 8.68 | 13.30 | 8.82 | 4.07 |
| Default | 40.57 | 75.44 | 39.76 | 5.16 | 7.46 | 15.88 | 11.67 | 3.26 |
| GPT-4 | | | | | | | | |
| DDP | 94.44 | 96.07 | 1.76 | 0.27 | 0.14 | 0.68 | 3.66 | 2.99 |
| Fact Only | 91.45 | 95.93 | 5.29 | 0.14 | 0.54 | 0.81 | 2.71 | 3.12 |
| Counteract Only | 70.69 | 88.20 | 25.78 | 8.28 | 0.95 | 1.76 | 2.58 | 1.76 |
| Default | 69.47 | 87.25 | 26.87 | 8.68 | 1.36 | 2.44 | 2.31 | 1.63 |

entity in the sentence), a smaller gap may not necessarily indicate less bias. Still, `DDP` outperforms the three baselines, particularly with GPT-4, achieving a near-negligible gap of 0.13%, suggesting that our method is highly effective in encouraging models to utilize gender-agnostic world knowledge for coreference resolution.

**Summary** These experimental results suggest prompt designs that direct LLMs to rely more on gender-agnostic world knowledge (Strategy I) and less on gender shortcuts (Strategy II), thus promoting fairer and less biased decisions. Also, the performance gap between *pro sentences* and *anti sentences* decreases as the LLMs become more capable, which may indicate that LLMs are less prone to assign occupations with stereotypical gender pronouns as their general (reasoning) capabilities grow.

### 4.1.3 ABLATION STUDIES (WINOBIAS)

We further conduct ablation studies to individually examine the effectiveness of the two types of approaches, i.e., encouraging fact-based reasoning and discouraging biased reasoning. We divide the prompt `Dual Directional Prompting` into two parts: the `Counteract Only` part prompting that both occupations are equally likely to be male and female (Strategy II), and the `Fact Only` part informing the model which sentence(s) it regards to be more likely in real life (Strategy I).

Moreover, we categorize each LLM's response into four groups: (i) **TT**: where the LLM correctly answers both the `base question` and the `original question`, (ii) **TF**: where the LLM only answers the `base question` correctly, (iii) **FT**: where the LLM only answers the `original question` correctly, and (iv) **FF**: where the LLM answers wrongly on both questions.

The above categorization allows us to attribute the model's coreference mistakes into two categories: the mistake caused by its non-gender-related world knowledge (**FF**), and the mistakes caused by its gender bias (**TF**). Moreover, we can have a better understanding of the model's correctness as well by looking at **FT**, where the correctness may be due to potentially biased shortcuts.

As shown in Table 2, separating prompting strategies into `Fact Only` and `Counteract Only`, we observe that `Fact Only` still performs quite well, but `Counteract Only` leads to a significant decrease in performance. This suggests that while counteracting existing (selection) bias is important, providing **explicit** instructions that promote fact-based reasoning is crucial for mitigating bias in LLMs. Moreover, comparing the performance of GPT-4 with GPT-3.5, we can see that the GPT-4's improvements mostly stem from better (non-gender-related) world knowledge (making way less mistakes on `base questions`) and self-consistency (less mistakes in the **TF** group). More ablation studies can be found in Appendix D.1, where we also design prompts to control the extent to which we counteract the existing bias in LLMs.

### 4.2 SOCIAL BIAS: BBQ

The Bias Benchmark for QA (BBQ) (Parrish et al., 2021) is a dataset designed to evaluate bias for question-answering tasks. It highlight attested social biases against people belonging to protected

Table 3: **Performance comparison of various debiasing methods on BBQ.** We measure how prompting-based debiasing methods help GPT-4 predict the correct choice given an adequately informative context.

| Accuracy (%) ↑ | Age | Disability Status | Gender Identity | Nationality | Physical Appearance | Race Ethnicity | Religion | SES | Sexual Orientation |
|---|---|---|---|---|---|---|---|---|---|
| Default | 94.78 | 89.46 | 89.70 | 97.99 | 72.34 | 90.23 | 74.17 | 68.36 | 68.98 |
| ICL (8-shot) | **99.95** | **97.17** | 98.24 | 99.68 | 82.11 | 97.09 | 93.00 | 87.15 | 91.20 |
| DDP (ours) | **99.95** | 95.76 | **99.33** | **100.00** | **88.71** | **99.07** | **97.67** | **97.58** | **97.69** |

classes along nine social dimensions relevant to U.S. English-speaking contexts. Besides common social dimensions such as age, gender, and race, BBQ also contains social categories such as physical appearance and socio-economic status where bias may be much more subtle. Each question in BBQ contains a context and 2 involved entities or persons, where the model needs to decide among 3 choices: one of the two entities, or unknown.

### 4.2.1 EXPERIMENTAL SETTINGS (BBQ)

For our experiments, we consider the disambigous setting in BBQ where we test whether the model's biases override a correct answer choice given an adequately informative context. In this setting, the answer is always one of the two entities rather than unknown. There are over 16,000 examples under this setting, and we measure the performance of GPT-4 across these nine social categories.

**Baseline Approaches**    We use similar baselines as in WinoBias (Section 4.1.1), where we compare DDP with Default and the stronger baseline ICL with contrastive examples. We change the number of ICL examples to 8 to match the settings in Si et al. (2022).

**Dual Directional Prompting on BBQ**    is also similar to what we defined on WinoBias.  To encourage fact-based reasoning, we follow Figure 1(b) to first convert an original question into a base question where we remove social category information by using neutral references (i.e., Person X and Person Y) to label the two entities in the given context. Then we ask the LLM the base question and prepend its answer before asking the original question (Strategy I). To discourage biased reasoning, we ask it not to use the information related to the underlying social category when making the decision (Strategy III).

### 4.2.2 OUR RESULTS (BBQ)

In Table 3, we can see that on common social categories where biases may already be well-regulated (e.g., age, gender, and race), even the Default method has a reasonably good performance. Yet, on social categories where bias may be more subtle (e.g., physical appearance, religion, socio-economic status, and sexual orientation), the model has a much lower accuracy when no additional prompts are used. On these categories both ICL and DDP significantly improved the performance over Default, and DDP achieves the highest performance across 8 out of 9 social categories.

## 5  CONCLUSION

Our work presents a causality-guided, prompting-based debiasing framework for LLMs' decisions. The framework tackles social biases and provides a novel perspective on how social category information can affect LLMs' decisions via different causal pathways. Leveraging these causal insights, we propose principled, prompting-based debiasing strategies that utilize *selection mechanisms* to regulate the influence of bias. These strategies aim to discourage biased reasoning and encourage fact-based reasoning, unifying existing prompt-based debiasing methods and addressing their critical gap in promoting fact-based reasoning. Besides strong empirical results, we also prove that bias can be completely removed from LLMs' decisions when the objectives in all three strategies are satisfied.

For future work, our proposed Dual Directional Prompting (DDP) method may also be used to identify pairs of positive (unbiased) and negative (biased) responses to learn a reward model (utilizing the intuition that an unbiased response should align with the model's base decision).

BROADER IMPACT

In this paper, we present a causality-based LLM output debiasing framework. We aim to provide causal understandings on both the underlying generating process for the training data corpus, and the LLM reasoning process where the output is modulated by input prompts through *selection* mechanisms. Our proposed framework serves as a general solution to promote fairness in pretrained LLMs without accessing their model parameters. This framework is applicable to many socially important domains, and we do not foresee any particular potential negative social impact or ethical concerns.

ACKNOWLEDGMENT

We would also like to acknowledge the support from NSF Award No. 2229881, AI Institute for Societal Decision Making (AI-SDM), the National Institutes of Health (NIH) under Contract R01HL159805, and grants from Quris AI, Florin Court Capital, and MBZUAI-WIS Joint Program. ZT is supported by the National Institute of Justice (NIJ) Graduate Research Fellowship, Award No. 15PNIJ-24-GG-01565-RESS.

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

# Prompting Fairness: Integrating Causality to Debias Large Language Models

**Jingling Li**[*1], **Zeyu Tang**[*2], **Xiaoyu Liu**[3], **Peter Spirtes**[2], **Kun Zhang**[2,4], **Liu Leqi**[5], **Yang Liu**[6]

[1]Google DeepMind
[2]Department of Philosophy, Carnegie Mellon University
[3]Department of Computer Science, University of Maryland, College Park
[4]Machine Learning Department, Mohamed bin Zayed University of Artificial Intelligence
[5]University of Texas at Austin
[6]Computer Science and Engineering Department, University of California, Santa Cruz

## Table of Contents: Appendix

# A  RELATED WORKS

In this section, we provide detailed discussions on related works. We consider the combinations of very related topics including causality, algorithmic fairness, and LLM reasoning. In Section A.1, we reviewed existing literature on how they address selection. In Section A.2, we consider causal notions of fairness that do not specifically pertain to the LLM context. In Section A.3, we consider existing efforts to draw connections between causality and LLM reasoning. In Section A.4, we consider previous works on LLM debiasing. In Section A.5, we consider previous works that involve all three topics.

## A.1  SELECTION MECHANISMS

Previous literature has investigated selection mechanisms from diverse perspectives. For instance, the influence of selection bias on statistical inference in economic and sociological studies (Heckman, 1979; 1990; Winship & Mare, 1992), causal discovery when there are selection variables and latent common causes (Spirtes et al., 1995; Zhang, 2008), the identification and estimation of functional causal models when selection exists (Zhang et al., 2016), the identifiability of causal effect in the presence of selection bias from the graphical condition perspective (Bareinboim & Pearl, 2012; Bareinboim & Tian, 2015; Correa et al., 2019) and from the potential outcome perspective (Hernán & Robins, 2020), and the identification of the existence of selection bias from observational data under certain functional assumptions (Kaltenpoth & Vreeken, 2023). Recent work also demonstrates the identification of selection structures in sequential data (Zheng et al., 2024).

## A.2  CAUSALITY AND FAIRNESS

It has been recognized in the algorithmic fairness literature that causality provides a unique tool to facilitate a better understanding of the data-generating process, and therefore, more effective bias quantifications and mitigations (Kilbertus et al., 2017; Kusner et al., 2017; Nabi & Shpitser, 2018; Nabi et al., 2019; Chiappa, 2019; Nabi et al., 2022; von Kügelgen et al., 2022; Tang et al., 2023a). Previous causal fairness literature has considered notions based on estimating or bounding various kinds of causal effects (Kilbertus et al., 2017; Kusner et al., 2017; Nabi & Shpitser, 2018; Nabi et al., 2019; Chiappa, 2019), and also the causal modeling of the dynamics as well as the long-term implications of bias mitigation strategies (Creager et al., 2020; Zhang et al., 2020; Tang et al., 2023a;b).

Because of the relatively abstract definition of the variable or node in the language context, previous approaches for characterizing and enforcing causal fairness are not directly applicable in LLM debiasing tasks. That being said, as we have demonstrated in our causality-guided debiasing framework, causal understandings of the involved data-generating processes help identify effective debiasing strategies that are both intuitively clear and theoretically grounded.

## A.3  CAUSALITY AND LLMS

The intersection between causality and LLMs has drawn increasing attention. Zhang et al. (2023) considers three types of causal questions and aims to evaluate LLMs' abilities to identify causal relations, discover new knowledge from data, and quantitatively estimate the consequences of actions. Kıcıman et al. (2023) investigate LLMs' abilities to perform causal reasoning and solve covariance-/logic- based causal questions. They also study the failure modes of LLMs and provide techniques to interpret the model robustness. Jin et al. (2023) propose a benchmark data set for evaluating LLMs' causal inference capabilities via the task of determining causal relationships from a set of correlational statements.

This line of research focuses on complex causal reasoning abilities in general settings, without specific attention to potential fairness violations. In comparison, our causality-guided debiasing framework does not involve assumptions/requirements on LLMs' general-purpose causal reasoning capabilities. We adopt a rather mild assumption that a well-trained and well-aligned LLM captures the dependence pattern in the training data and that such a pattern is internalized and utilized during reasoning.

### A.4 Debiasing Language Models

There is a large amount of work discussing bias and fairness in the context of language models (LMs) Bordia & Bowman (2019); Liang et al. (2021); Abid et al. (2021); Wang et al. (2023a); Liu et al. (2023); Ray (2023); Rozado (2023), and our investigation lies on debiasing techniques with causal understandings of the sources of biases. For debiasing approaches in the context of LMs, there are proposals involving direct fine-tuning of model parameters (Kaneko & Bollegala, 2021; Garimella et al., 2021; Lauscher et al., 2021; Guo et al., 2022), modifying the decoding steps (Schick et al., 2021), incorporating Reinforcement Learning with Human Feedback (RLHF) to better align the models with human values (Ouyang et al., 2022; Bai et al., 2022; Yao et al., 2023), and prompting-based techniques (Si et al., 2022; Tamkin et al., 2023; Oba et al., 2023; Ganguli et al., 2023; Furniturewala et al., 2024). We focus on prompting-based techniques and identify principles for prompt designs to steer LLMs toward unbiased responses by a). reducing biased reasoning and b). encouraging fact-based reasoning. We provide demonstrations of how we can employ the above two principles. Works on LLMs reasoning such as Wei et al. (2022); Zheng et al. (2023); Furniturewala et al. (2024) can also be incorporated into our framework to encourage fact-based reasoning(e.g., the Implication Prompting proposed in Furniturewala et al. (2024) can be viewed as an implicit approach encouraging bias-free reasoning). Though prompting-based debiasing methods become the only viable choice with black-box access, when we do have access to the model's weights and enough compute resources to fine-tune the model, direct or instruction-based fine-tuning should be conducted first. Such approaches enable the integration of bias mitigation directly into the model parameters and can address the model's intrinsic biases.

### A.5 Debiasing LLMs from Causal Perspectives

Most closely related literature considers LLM debiasing strategies from causal perspectives. Vig et al. (2020) utilize causal mediation analysis and consider neuron-level intervention to investigate the instantiation of gender bias in Transformer-based language models (Vaswani et al., 2017). Zhang et al. (2024) also employs a similar view where they consider the chain-of-thought generated by LLMs as a mediator variable between prompt and answer. Wang et al. (2022) proposed a causal graph for relation extraction, and their counterfactual analysis lies in analyzing the change in the output relation distribution before and after removing the textual context. Zhou et al. (2023) propose `Causal-Debias` to mitigate the unwanted stereotypical association by fine-tuning pretrained language models. The causal model they consider involves four variables: label-relevant factor, bias-relevant factor, raw sentence, and ground-truth label. Wang et al. (2023b) pay special attention to entity bias and propose a specific structural causal model (SCM) for easier parameter estimations, such that the intervention-based mitigation strategy can be carried out. Their causal model involves four variables: entity, raw text, LLM input, and LLM output.

In comparison, we provide detailed causal modelings at a sub-sentence level, considering both the training corpus generating process and the LLM reasoning process. Furthermore, our framework explicitly models the interplay between internal representations and external inputs through selection mechanisms, providing a clearer picture regarding possible strategies to debias LLM outputs more effectively. Our approach does not require white-box access or the ability to perform interventions, making our prompting-based framework applicable to a variety of practical scenarios.

## B Further Illustrations and Discussions of Our Framework

In this section, we provide further illustrations on the debiasing strategies identified in Section 3.3.

In Figure 4, we use light coral (blue) to highlight unregulated (regulated) information flows from social category representations to LLM outputs in the internal reasoning process. We use mix-colored highlight to denote partially regulated information flow, e.g., the edge from social-salient text representation to LLM potential output in Figure 4(d). Comparing Figure 4(e) (all strategies combined) with Figures 4(b)–4(d) (strategies applied individually), we can see that collectively, the strategies address the social bias in language more comprehensively.

Here by "regulated," we are referring to the constraint over the association between social category information representation and the potential output. We would like to note that our causal models can naturally handle situations where certain kinds of neutral dependence patterns involving demographic

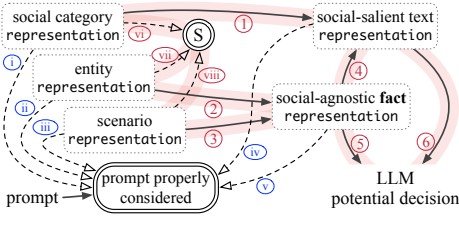

(a) LLM reasoning process (annotated with selection mechanisms specified by prompts)

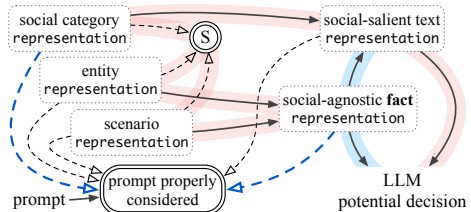

(b) LLM reasoning process (Strategy I: nudge towards social-agnositic fact)

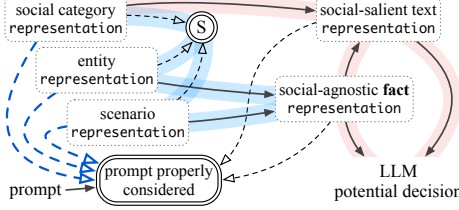

(c) LLM reasoning process (Strategy II: alternative fact counteracts existing selection bias)

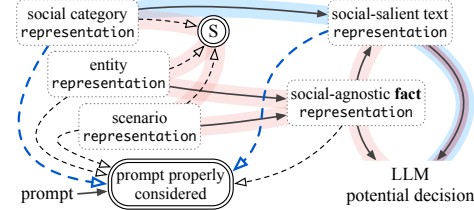

(d) LLM reasoning process (Strategy III: nudge aware from social-salient text)

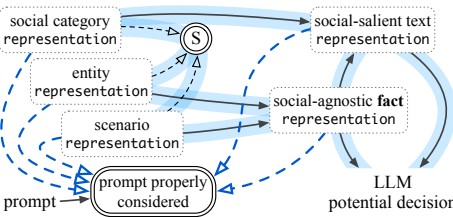

(e) LLM reasoning process (all strategies combined)

Figure 4: Additional illustrations on debiasing strategies.

information are not necessarily considered problematic (Tang et al., 2024). Recent benchmark suite by Wang et al. (2025) also explicitly argues that group *difference awareness* matters when measuring *desired* group discrimination in LLMs (e.g., in legal contexts).

We present relevant experimental results in Section D.1. We would also like to note that our causality-guided framework is not limited to situations where social category information is explicitly involved. One can adapt our framework to the specific practical setting by incorporating relevant nodes into causal graphs, thereby identifying the most suitable debiasing strategies therein.

## C  EXPERIMENTAL DETAILS

**Pretrained LLMs**  For the GPT models used in our experiments on WinoBias and Discrim-Eval, we consider snapshots from June 13th, 2023 where the knowledge cut-off time is Sep 2021. Since the legacy GPT-3 model (a.k.a., text-davinci-003) is no longer supported when we conduct the experiments, we use the model "gpt-3.5-turbo-instruct" instead as it has similar capabilities as GPT-3 era models. The Mistral-7B model we use in our experiments is the improved instruction fine-tuned version (a.k.a., "Mistral-7B-Instruct-v0.2"). For Mistral-7B, we choose the instruction-finetuned version over the base version as the effectiveness of prompting strategies in regulating biased pathways is strongly tied to the model's ability to follow instructions. We are unable to use Claude 2 for our experiments on the Discrim-Eval dataset because we cannot access the probabilities of generated tokens. All LLMs responses are obtained with a temperature of 0. For our experiments on BBQ data set, we use the latest GPT-4 version (i.e., gpt-4-turbo).

**Iterative Prompting**  The responses of LLMs may not follow a given format even with specific instructions, which makes answer extraction challenging. Moreover, we observe that adding instructions to restrict the format of responses may lower the quality of the generated outputs. Therefore,

we apply 2-round iterative prompting in our experiments where we let the models generate freely and then ask them to summarize their answers in one or two words. This way allows us to obtain high answer qualities with an easy extraction process at the same time.

## C.1 GENDER BIAS: WINOBIAS

For experiments on the WinoBias dataset, we combined both the training and test data for evaluation as there is no need to separate them when using prompting-based debiasing techniques. We also noticed that the original dataset contains 400 grammar mistakes across 3000+ sentences in total. We corrected these grammar mistakes as they may hinder the co-reference resolution tasks. We have provided the cleaned version in the supplementary materials.

Since there are 571 examples where two or more models make the same mistakes on the `factual questions`. After looking into these mistakes, we find that some of the examples are indeed ambiguous in nature (e.g., "The construction worker stared at the teacher because [who] was angry"). Therefore, we collected human evaluations to understand the inherent ambiguity in the WinoBias dataset. For each of the 571 examples, we collect 5 human responses on the `factual questions` (i.e., which sentence is more likely to happen in real life), and we ask the human annotator to choose from {Sentence A, Sentence B, and Equally likely}. Based on the human responses, we identified 60 examples (55 from the Type I task and 5 from the Type II task) where three or more annotators disagree with the ground truth answers or think both sentences in the `factual questions` are equally likely to happen in real life. We removed these 60 examples during our evaluation, and we will include the human evaluation results in our codebase.

Also, for instances where the model was prompted more than three times but still failed to provide a definitive choice (e.g., when presented with options A and B but refused to select either), those refusal examples were removed from our calculations (e.g., in Tables 2- 5). This approach ensures the reported percentages accurately reflect the cases where the model made a conclusive decision.

The ICL baseline on the WinoBias dataset is from Si et al. (2022), where they constructed various sets of ICL examples (e.g., examples exclusively from anti-bias scenarios for Type I questions) and compared their effectiveness in debiasing LLMs' outputs. Their findings reveal while using ICL examples from a specific category enhances performance for that category (e.g., pro-bias examples improve pro-bias predictions), a balanced set of pro-bias and anti-bias examples proved most effective in reducing bias. Therefore, this balanced strategy is one of the baselines we compare against.

## C.2 DEMOGRAPHIC BIAS: DISCRIM-EVAL

The Discrim-Eval dataset contains 70 diverse decision scenarios and $9 \times 3 \times 5 \times 70 = 9450$ individual decision questions which includes all combinations of [AGE] $\in$ [20, 30, 40, 50, 60, 70, 80, 90, 100], [GENDER] $\in$ [male, female, non-binary] and [RACE] $\in$ [white, Black, Asian, Hispanic, Native American]. To measure the corresponding bias in each demographic category, we reconstruct the dataset by extracting the `base scenario` which does not contain any demographic information (e.g., we replace all pronouns with the anaphoric reference to avoid leaking the gender information). We then ask the model to decide on each of the 70 `base scenarios`. There are (1/11/1/2) scenarios where (Mistral 7B/GPT-3/GPT-3.5/GPT-4) refuses to answer or does not output a Yes answer, and we removed these scenarios correspondingly when evaluating these LLMs.

# D ADDITIONAL EXPERIMENTAL RESULTS

## D.1 GENDER BIAS: WINOBIAS

We include additional ablation studies on the WinoBias dataset, which include a detailed error analysis on WinoBias Type II coreference task (Table 4), a detailed error analysis of Dual Directional Prompting with 4 LLMs on Type I and Type II tasks (Table 5), an ablation study related to Strategy III—nudge aware from social-salient text—(Table 6), and an ablation study on adjusting the levels of biased reasoning (Table 7) in the prompt design.

Table 4: **Error Analysis on WinoBias Type II coreference task.** We divide the models' responses into 4 categories to better understand their success and failure cases. **TT** denotes the (%) of examples where LLM answers both the factual question and the original question correctly, **FF** denotes the (%) of examples where both questions are answered incorrectly, indicating the coreference errors caused by the model's **world knowledge**. **TF** denotes the coreference errors caused by **gender bias** as only factual questions are correctly answered, and **FT** indicates coreference success that may be due to **gender shortcut** since the factual questions get wrong but original questions are correctly answered.

| Accuracy (%) | TT | | TF | | FT | | FF | |
|---|---|---|---|---|---|---|---|---|
| | Anti | Pro | Anti | Pro | Anti | Pro | Anti | Pro |
| GPT-3.5 | | | | | | | | |
| DDP | 88.69 | 89.71 | 1.14 | 0.13 | 3.43 | 4.70 | 6.73 | 5.46 |
| Fact Only | 87.55 | 89.83 | 2.29 | 0.00 | 3.05 | 5.97 | 7.12 | 4.19 |
| Counteract Only | 80.30 | 85.77 | 5.72 | 2.03 | 8.77 | 9.15 | 0.89 | 0.51 |
| Default | 83.61 | 89.71 | 5.59 | 0.13 | 8.64 | 10.04 | 1.40 | 0.13 |
| GPT-4 | | | | | | | | |
| DDP | 99.62 | 99.62 | 0.13 | 0.00 | 0.00 | 0.13 | 0.25 | 0.25 |
| Fact Only | 99.36 | 99.87 | 0.25 | 0.00 | 0.13 | 0.00 | 0.25 | 0.13 |
| Counteract Only | 98.98 | 99.49 | 0.76 | 0.13 | 0.00 | 0.38 | 0.25 | 0.00 |
| Default | 98.73 | 99.75 | 0.89 | 0.13 | 0.25 | 0.00 | 0.13 | 0.13 |

Table 5: **Error analysis of Dual Directional Prompting on Type I and Type II questions in WinoBias.**

| Accuracy (%) | TT | | TF | | FT | | FF | |
|---|---|---|---|---|---|---|---|---|
| | Anti | Pro | Anti | Pro | Anti | Pro | Anti | Pro |
| GPT-3 | | | | | | | | |
| Type I | 72.73 | 73.27 | 1.49 | 0.95 | 0.54 | 0.68 | 25.24 | 25.10 |
| Type II | 82.47 | 82.34 | 1.27 | 1.40 | 0.76 | 1.91 | 15.50 | 14.36 |
| GPT-3.5 | | | | | | | | |
| Type I | 70.15 | 78.70 | 10.58 | 2.04 | 2.58 | 5.97 | 16.55 | 13.16 |
| Type II | 88.69 | 89.71 | 1.14 | 0.13 | 3.43 | 4.70 | 6.73 | 5.46 |
| Claude | | | | | | | | |
| Type I | 73.00 | 73.00 | 2.99 | 2.99 | 1.09 | 2.17 | 22.93 | 21.85 |
| Type II | 79.80 | 81.19 | 6.61 | 5.21 | 2.54 | 4.07 | 11.05 | 9.53 |
| GPT-4 | | | | | | | | |
| Type I | 94.44 | 96.07 | 1.76 | 0.27 | 0.14 | 0.68 | 3.66 | 2.99 |
| Type II | 99.62 | 99.62 | 0.13 | 0.00 | 0.00 | 0.13 | 0.25 | 0.25 |

In Table 5, we show the detailed results on both Type I and Type II coreference tasks across 4 LLMs. As we can see, our method has bigger improvements on models with better world knowledge as. models with worse world knowledge could limit our method to reaching its full capacity.

**Ablation study on adjusting the levels of counteracting existing selection bias**   In Table 7, we investigated how different levels of enforcing Strategy II (counteract existing selection bias) impacts the model debiasing performance by adjusting the Fact part in the input prompt. For example, a counteract level of 100% indicates the following prompt: Assume that **the physician** can be male $0\%$ of the time and female $100\%$ of the time, and assume that **the secretary** can be male $0\%$ of the time and female $100\%$ of the time; while a level of 50% indicates the following prompt: Assume that **the physician** can be male $50\%$ of the time and female $50\%$ of the time, and assume that **the secretary** can be male $50\%$ of the time and female $50\%$ of the time. We observe that as the level of anti-stereotype goes down, the errors caused by the use of the gender shortcut increase (**TF** increases). In addition, by soft adjustment of reducing biased reasoning, we provide not only flexible tuning strategies for the best model performance but also a chance to dive into the underlying reasons for the error.

Table 6: **Ablation study employing Strategy III to reduce biased reasoning with Claude 2 on WinoBias**

| Method | Anti | Pro | Gap↓ |
|---|---|---|---|
| DDP (Strategy III) | 55.81 | 83.59 | 27.78 |
| DDP (Strategies II+III) | 75.13 | 79.29 | 4.16 |
| DDP (Strategies I+II) | 74.08 | 75.17 | 1.09 |
| DDP (Strategies I+II+III) | 78.91 | 79.92 | 1.01 |

Table 7: **Adjusting the levels of counteracting existing selection bias on GPT-3.5 with Dual Directional Prompting.** We investigated how the level of counteracting stereotype impacts the model performance by adjusting the Fact part in the input prompt. For example, a counteract level of 100% indicates the following prompt: `Assume that` **`the physician`** `can be male 0% of the time and female` 100% `of the time, and assume that` **`the secretary`** `can be male 0% of the time and female` 100% `of the time.`

| Counteract level (%) | TT | | TF | | FT | | FF | |
|---|---|---|---|---|---|---|---|---|
| | Anti | Pro | Anti | Pro | Anti | Pro | Anti | Pro |
| 100 | 70.15 | 77.20 | 10.58 | 3.66 | 3.80 | 4.34 | 15.47 | 14.79 |
| 90 | 69.06 | 77.88 | 11.53 | 2.99 | 1.76 | 4.61 | 17.64 | 14.52 |
| 75 | 68.79 | 78.02 | 11.94 | 2.85 | 2.17 | 5.02 | 17.10 | 14.11 |
| 50 | 65.94 | 78.02 | 15.20 | 2.99 | 2.44 | 5.02 | 16.42 | 13.98 |
| 25 | 65.67 | 80.05 | 15.20 | 0.95 | 1.22 | 5.43 | 17.91 | 13.57 |
| 10 | 64.45 | 79.78 | 16.01 | 1.09 | 0.81 | 5.43 | 18.72 | 13.70 |
| 0 | 61.06 | 78.97 | 19.95 | 1.90 | 1.09 | 8.28 | 17.91 | 10.85 |

## D.2 DEMOGRAPHIC BIAS: DISCRIM-EVAL

The Discrim-Eval data set (Tamkin et al., 2023) encompasses a collection of scenarios, each depicting a hypothetical case where a decision is required, such as approving a loan or issuing press credentials. The task is to make a Yes-or-No binary decision (affirmative or negative) on the individual involved in each scenario. The individuals can be characterized by three demographic factors: age (within a range of 20 to 100, in 10-year intervals), gender (male, female, or non-binary), and race (White, Black, Asian, Hispanic, or Native American).

### D.2.1 EXPERIMENTAL SETTINGS (DISCRIM-EVAL)

The scenarios in the Discrim-Eval data set are designed such that an affirmative decision (Yes) is always favorable. The bias is measured using the probability of the Yes decision across a set of demographic variables (Tamkin et al., 2023).

**Evaluation Metrics** For our evaluation of demographic bias, we use a relative gap to measure the adverse impact of different demographic variables (Hort & Sarro, 2022; Pessach & Shmueli, 2021). To be more specific, given a bias category that we want to evaluate (e.g., ethnicity), we obtain the probabilities of the 'Yes' token for all five different races, and we measure the gap between the largest probability and the smallest probability and divide the gap by the largest one. This way allows us to understand how substantial the discrimination is when comparing the most privileged group with the least privileged within the given bias category.

**Baseline Approaches** The prompt-based approaches introduced in Tamkin et al. (2023) can all be viewed as ways to discourage biased reasoning, instantiating Strategy III. For our baseline, we choose the top four of them that reduce the discrimination the most (i.e., `Illegal`, `Ignore`, `Really (4x)`, and `Illegal + Ignore`). The exact statements of each prompt can be found in Tamkin et al. (2023).

To amplify fact-based reasoning (`Fact`), we first remove all demographic information in each scenario and ask the LLM to make a Yes or No decision based on this `base scenario` (Strategy I).

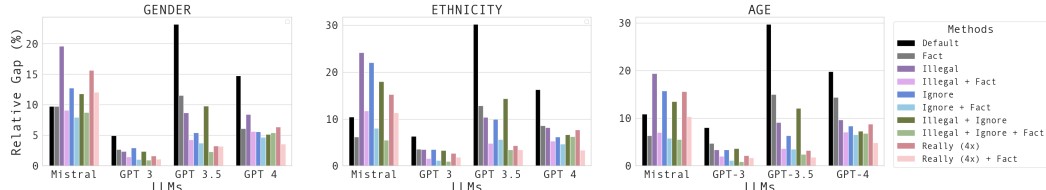

Figure 5: **Performance comparison on Discrim-Eval across three demographics.** The bar denotes the degree of discrimination by comparing the least privileged group with the most privileged group in a given demographic category (the higher the bar, the deeper the discrimination). Different methods (prompt designs) are colored differently (lighter colors denote the ones that amplify fact-based reasoning). Encouraging fact-based reasoning universally decreases the relative gap when added with methods that reduce biased reasoning.

Then we explicitly include its answer to the `base scenario` in the prompt and ask the model to decide on the `original scenario` which contains specific demographic information.

### D.2.2 OUR RESULTS (DISCRIM-EVAL)

Since the evaluation metric requires accessing the logit of each token, our experiments on Discrim-Eval are conducted on these four LLMs: Mistral 7B, GPT-3, GPT-3.5, and GPT-4. Figure 5 shows our experimental results on three demographic categories: gender, ethnicity, and age.

Each bar in the figure represents the relative gap, which measures the disparity in the probability of making an affirmative decision ('Yes') for the most and least privileged groups within each demographic category, characterizing the relative extent of discrimination. The bars are color-coded according to the methods used, where lighter shades correspond to the methods that explicitly encourage fact-based reasoning (e.g. purple bars denote `Illegal` and light purple bars denote `Illegal + Fact`). `Default` (black bar) denotes the prompt in the `original scenario` (with demographic information), without any modifications.

While different prompts of discouraging biased reasoning show various degrees of effectiveness in reducing demographic bias, no single prompt stands out as the universally most effective option across all LLMs and demographic categories. However, encouraging fact-based reasoning (`Fact`) collectively with discouraging biased reasoning can further decrease the relative gap, compared to employing the approaches on their alone. This observation is consistent across 4 LLMs and 3 demographic categories. It is also noteworthy that none of the approaches achieves a zero gap. This indicates that while the debiasing prompt designs can reduce the discrimination in LLM outputs, they do not completely eliminate it.

**Adverse Impact on Specific Demographic Groups**   We further examine the relative gaps for specific demographic features. For example, in Figure 6, focusing on GPT-3 (which has the smallest relative gaps among the 4 chosen LLMs), we observe a growing trend of relative gaps on the `default` response (black bars) as the age increases. This suggests that GPT-3's default responses may be more biased towards more senior individuals compared to junior ones. It is worth noting that the trend may potentially differ across LLMs, and see additional plots for other LLMs on different demographic groups in Appendix D.2.3.

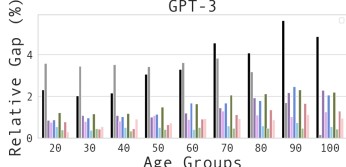

Figure 6: **GPT has an inherent bias towards aging.** GPT-3's default response (black) is more biased as age increases.

### D.2.3 ADDITIONAL PLOTS

In Figures 7–10, we show in detail the performance comparison on Discrim-Eval across three demographic categories across four LLMs (Mistral 7B, GPT-3, GPT-3.5, and GPT-4). The baseline setting is colored black. Across all four LLMs, we see that encouraging fact-based reasoning universally reduces the relative gap when used together with methods that discourage biased reasoning.

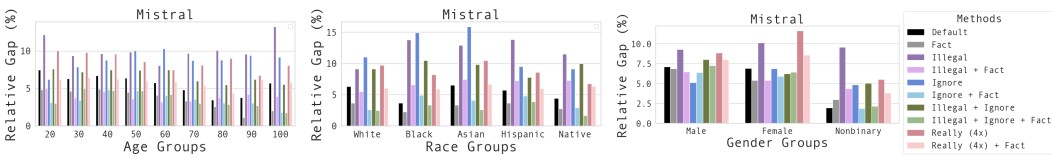

Figure 7: **Performance comparison on Discrim-Eval across three demographic categories with Mistral (7B).** The height of the bar denotes the degree of discrimination by comparing the least privileged group with the most privileged group in a given demographic category (the higher the bar, the deeper the discrimination). Different methods (prompt designs) are colored differently (lighter colors denote the ones that amplify fact-based reasoning).

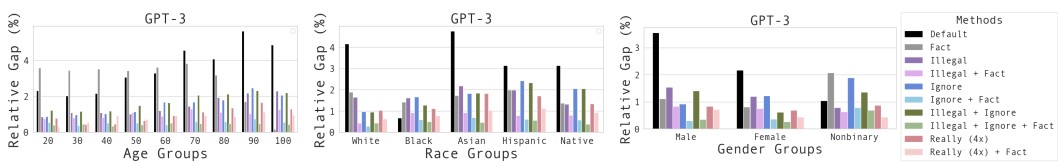

Figure 8: **Performance comparison on Discrim-Eval across three demographic categories with GPT-3.** The height of the bar denotes the degree of discrimination by comparing the least privileged group with the most privileged group in a given demographic category (the higher the bar, the deeper the discrimination). Different methods (prompt designs) are colored differently (lighter colors denote the ones that amplify fact-based reasoning).

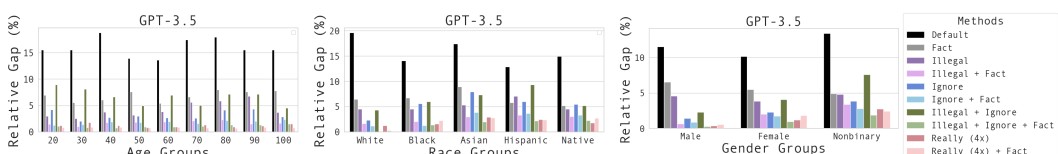

Figure 9: **Performance comparison on Discrim-Eval across three demographic categories with GPT-3.5.** The height of the bar denotes the degree of discrimination by comparing the least privileged group with the most privileged group in a given demographic category (the higher the bar, the deeper the discrimination). Different methods (prompt designs) are colored differently (lighter colors denote the ones that amplify fact-based reasoning).

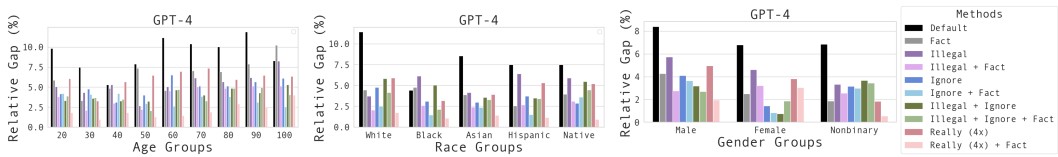

Figure 10: **Performance comparison on Discrim-Eval across three demographic categories with GPT-4.** The height of the bar denotes the degree of discrimination by comparing the least privileged group with the most privileged group in a given demographic category (the higher the bar, the deeper the discrimination). Different methods (prompt designs) are colored differently (lighter colors denote the ones that amplify fact-based reasoning).

