# OpenReview forum: "Prompting Fairness: Integrating Causality to Debias Large Language Models"
_ICLR.cc/2025/Conference — ICLR 2025 Poster_

### Official Review · Reviewer_LwQ1 · 2024-10-29

**Soundness:** 3
**Presentation:** 3
**Contribution:** 3
**Rating:** 6
**Confidence:** 4

**Summary:**

This article conducted a causality analysis for bias in LLM's decision and provided a prompting-based solution for bias mitigation. The solution included three strategy pathways and demonstrated that the combining strategy can achieve comprehensive debiasing.

**Strengths:**

The given causality-analysis clarified the origins of bias in well-trained LLMs, making the proposed bias mitigation strategies more explainable.

In addition to examining the influence of training data corpus and prompts, the authors also considered bias caused by data selection, offering a more systematic and comprehensive bias-mitigation solution.

**Weaknesses:**

1. The assumption in A.3 CAUSALITY AND LLMS,
"We adopt a rather mild assumption that a well-trained and well-aligned LLM captures the dependence pattern in the training data and that such a pattern is internalized and utilized during reasoning"
needs to be put in the main text. If LLMs are not well-trained and well-aligned, then the author's prompting-based de-dependency method will no longer be so effective, so I think this is an assumption worth stating in the main text.

2. There is a lack of discussion related to other prompting-based debiasing methods in LLMs, such as those in [1][2][3].

[1] Zhang, C., Zhang, L., Zhou, D., & Xu, G. (2024). Causal Prompting: Debiasing Large Language Model Prompting based on Front-Door Adjustment. arXiv preprint arXiv:2403.02738.

[2] Li, J., Tang, Z., Liu, X., Spirtes, P., Zhang, K., Leqi, L., & Liu, Y. (2024). Steering LLMs Towards Unbiased Responses: A Causality-Guided Debiasing Framework. arXiv preprint arXiv:2403.08743.

[3] Furniturewala, S., Jandial, S., Java, A., Banerjee, P., Shahid, S., Bhatia, S., & Jaidka, K. (2024). Thinking Fair and Slow: On the Efficacy of Structured Prompts for Debiasing Language Models. arXiv preprint arXiv:2405.10431.

**Questions:**

1. When the three strategies are combined, both social-salient text representation and social-agnostic fact
representation are independent of social category
representation. Does this mean that it is equivalent to using only prompts Q:{Base Question}, a prompt that is completely unrelated to sensitive features, to let LLM complete the answer to Original Question? If not, when both social-salient text representation and social-agnostic fact
representation are independent of social category
representation, what is the performance difference of LLM decision brought by prompts Q:{Base Question} and Q:{Given that your answer to {Base Question} is {Answer to Base Question}, Assume male and female are equally represented in drivers and in teachers. Do not answer the question using gender information. {Original Question}}? Can the author give experimental results on this point?
2. Should there be a solid line in Figure 3 connecting prompt to LLM potential decision?
3. In addition to the fact that prompting-based techniques are suitable for dealing with black-box scenarios, can the authors add a discussion on the advantages and limitations of prompting-based techniques compared to direct fine-tuning of model parameters and prompting-based techniques, for example, a limitation of  prompting-based technique ——the need for human users to be proactive and knowledgeable to complete debiasing?
4. Can the authors provide a comparison of this work with other prompting-based debiasing methods mentioned in weakness, either experimentally or analytically?
5. In the Table 2, why the sum of the true-rate (42.33%, 32.02%) and false-rate (8.68%, 8.82%) of the base question under GPT-3.5/Counteract Only/Anti case is not 1?

---

> ### Author Response · Authors · 2024-11-30
> **Reply Part I to Reviewer LwQ1**
>
> Thanks for the detailed and thoughtful comments and questions, as well as the time and effort devoted! Please see our responses to each specific point below:
>
> ---
>
> ### **C1:** "The assumption in A.3 ... needs to be put in the main text. If LLMs are not well-trained and well-aligned, then the author's prompting-based de-dependency method will no longer be so effective, so I think this is an assumption worth stating in the main text."
>
> **A1:** We sincerely thank the reviewer for their insightful feedback. Following your suggestion, we have incorporated the assumption directly into the main text (lines 320–323). We truly appreciate your suggestion, which has significantly enhanced the presentation of our work.
>
> ---
>
> ### **C2:**  "There is a lack of discussion related to other prompting-based debiasing methods in LLMs, such as those in [1][2][3]."
>
> **A2:** Thank you for pointing out these references! We have incorporated discussions of these related works into our updated manuscript. Specifically, we contextualize how these prompting-based debiasing methods align with and differ from our causality-guided framework, further highlighting the distinct contributions of our approach.
>
> ---
>
> ### **C3:** "When the three strategies are combined, ... Does this mean that it is equivalent to using only prompts Q:{Base Question}, a prompt that is completely unrelated to sensitive features, to let LLM complete the answer to the Original Question? If not, ..., what is the performance difference of LLM decision brought by prompts Q:{Base Question} and Q:{Given that your answer to {Base Question} is {Answer to Base Question}, Assume male and female are equally represented in drivers and in teachers. Do not answer the question using gender information. {Original Question}}? Can the author give experimental results on this point?"
>
> **A3:** Thank you for your insightful questions!
>
> 1. Does this mean that it is equivalent to using only prompts Q:{Base Question}, a prompt that is completely unrelated to sensitive features, to let LLM complete the answer to the Original Question?
>
> Are you suggesting that using the LLM's answer to the "Base Question" directly as its answer to the "Original Question"? While this approach is possible (in this case we will not have any bias gaps since the Base Question is shared across related Original Questions), our **Fact Only** approach (Strategy I) deliberately includes the Original Question in the prompt format:
>
> `Given that your answer to {Base Question} is ..., {Original Question}?`
>
> This formulation ensures the model has complete context, which may help its decision-making. Simply relying on the Base Question’s response might neglect nuances in the Original Question that could influence the answer.
>
> 2. What is the performance difference of LLM decision brought by prompts Q:{Base Question} and Q:{Given that your answer to {Base Question} is {Answer to Base Question}, Assume male and female are equally represented in drivers and in teachers. Do not answer the question using gender information. {Original Question}}?
>
> Thank you for the suggestion! We have conducted an additional ablation study (now presented in Table 6 of our updated manuscript).  As we expected, combining all three strategies achieves slightly better performance than DDP wth just Strategy I and II. Interestingly, we observed that simply instructing the model to avoid using gender-related information (Strategy III) has limited impact in isolation. However, when used alongside the other strategies, it enhances the effectiveness of the debiasing framework. These results further support the effectiveness of our causality-guided debiasing framework.
>
> ---
>
> ### **C4:** "Should there be a solid line in Figure 3 connecting prompt to LLM potential decision?"
>
> **A4:** This is an excellent question! While it is tempting to use a direct link to connect prompt to LLM potential decision, the prompt actually does not serve as a direct cause of, nor a (hard or soft) intervention upon, the LLM potential decision. Instead, the input prompt directly changes the selection variable "prompt properly
> considered" (PPC) that regulates LLM potential decision via selection mechanisms. This distinction is discussed in detail in lines 224–233 of our paper. Please let us know if the content helps address the question.
>
> ---

---

> > ### Author Response · Authors · 2024-11-30
> > **Reply Part II to Reviewer LwQ1**
> >
> > ### **C5**: "In addition to the fact that prompting-based techniques are suitable for dealing with black-box scenarios, can the authors add a discussion on the advantages and limitations of prompting-based techniques compared to direct fine-tuning of model parameters and prompting-based techniques?"
> >
> > **A5:**  Thank you for the insightful suggestion! While fine-tuning large language models (LLMs) can help mitigate intrinsic biases, it is often prohibitively expensive and inaccessible for individual users or organizations. In contrast, prompting-based techniques provide a cost-effective and efficient alternative, especially when dealing with closed-source or black-box models as you've mentioned. However, when access to the model's weights and resources for fine-tuning is available, direct or instruction-based fine-tuning should be conducted first. Such approaches enable the integration of bias mitigation directly into the model’s parameters and can address the model's intrinsic biases. We have incorporated a detailed discussion of these advantages and limitations in the updated manuscript (Appendix A.4).
> >
> > ---
> >
> > ### **C6:**  "In the Table 2, why the sum of the true-rate (42.33%, 32.02%) and false-rate (8.68%, 8.82%) of the base question under GPT-3.5/Counteract Only/Anti case is not 1?"
> >
> > **A6:** This is an insightful observation. The values in Table 2 do not sum to 100% because certain examples were excluded from the calculations. Specifically, in instances where the model was prompted more than three times but still failed to provide a definitive choice (e.g., when presented with options A and B but refused to select either), those refusal examples were removed from our calculations in Table 2. This approach ensures the reported percentages accurately reflect the cases where the model made a conclusive decision. We have clarified this point in the revised manuscript as well (Appendix C.1).

---

> > > ### Author Response · Authors · 2024-12-01
> > >
> > > Dear reviewer LwQ1,
> > >
> > > As the discussion period ends tomorrow, we believe we have addressed all the questions and requested additional ablation studies in your initial review. Could you please clarify that? We are looking forward to your comments and feedback on our rebuttal. Thank you so much!

---

> > > > ### Author Response · Authors · 2024-12-02
> > > > **Discussion phase ending AOE today**
> > > >
> > > > Dear Reviewer,
> > > >
> > > > As the discussion phase quickly approaches an end, we are looking forward to your comments and feedback on our rebuttal. Thank you again for the time and effort!

---

### Official Review · Reviewer_qVWj · 2024-10-31

**Soundness:** 3
**Presentation:** 2
**Contribution:** 2
**Rating:** 6
**Confidence:** 2

**Summary:**

This paper proposes a causality-guided framework for debiasing large language models through prompting strategies. The authors introduce a novel perspective to identify how social information influences LLM decisions through different causal pathways and develop principled prompting strategies to regulate these pathways through selection mechanisms. The framework encompasses two main approaches: encouraging fact-based reasoning and discouraging biased reasoning. The authors validate their framework through extensive experiments on multiple benchmark datasets (WinoBias, BBQ, and Discrim-Eval) across various social dimensions, demonstrating its effectiveness in debiasing LLM decisions while maintaining strong performance.

**Strengths:**

This work presents a novel theoretical framework that bridges causality and LLM debiasing, providing clear intuitions and principled strategies for addressing bias. The causal modeling of both training data generation and LLM reasoning processes offers valuable insights into bias sources and mitigation approaches.

**Weaknesses:**

Although the authors demonstrate improved performance across multiple models, there's limited discussion of how the effectiveness of their approach might vary with model scale or architecture. Additionally, while the authors show reduced bias metrics, there could be more analysis of potential trade-offs between bias reduction and task performance.

**Questions:**

How does the effectiveness of the proposed debiasing strategies vary with model size and architecture?

---

> ### Author Response · Authors · 2024-11-30
> **Reply to Reviewer qVWj**
>
> Thanks for the detailed and thoughtful comments and questions, as well as the time and effort devoted! Please see our responses to each specific points below:
>
>
> ---
>
> ### **C1:** "while the authors show reduced bias metrics, there could be more analysis of potential trade-offs between bias reduction and task performance"
>
> **A1:** Thank you for the suggestions! We would like to clarify that the observed reduction in task performance does not indicate a trade-off between bias reduction and reasoning capability but rather represents **a recalibration of the model's reliance on biased shortcuts**.  The performance decrease aligns with the correction of LLMs' dependency on biased social stereotypes rather than a reduction in their reasoning ability.
>
>
> This claim is supported by our detailed ablation studies presented in Table 2. Specifically, when GPT-3.5 was asked base questions—neutral reformulations designed to remove gendered pronouns and test world-knowledge-based reasoning—it answered 19.13% incorrectly (19.13% is the sum of errors from "FT-Pro" and "FF-Pro" or equivalently "FT-Anti" and "FF-Anti" categories, e.g., 5.97% + 13.15%). However, However, when the original pro-stereotype questions associated with these same base questions were asked directly (Default), GPT-3.5 answered approximately 83% (15.88%/19.13%) of them correctly. This pattern indicates the model’s tendency to rely on biased gender shortcuts to "solve" questions.
>
> The DDP method reduced this reliance from 15.88% to 5.97%, illustrating that the observed decrease in accuracy on pro-stereotypical questions (from 94.03% to 84.67%, as shown in Table 1) reflects a mitigation of biased reasoning pathways, not a loss in intrinsic reasoning capacity. Crucially, the results show that DDP nudges the model to reason based on neutral factual knowledge rather than exploit socially biased cues.
>
> ---
>
> ### **C2:**  "How does the effectiveness of the proposed debiasing strategies vary with model size and architecture?"
>
> **A2:** This is an excellent point. For black-box models, we do not have access to details about their underlying size or architecture. However, as discussed in lines 454–457, our findings indicate that the performance gap between pro-stereotypical and anti-stereotypical sentences narrows as LLMs become more capable. This suggests that as LLMs enhance their general reasoning abilities, they may become less prone to associating occupations with stereotypical gender pronouns.
>
> For open-source models, such as the Mistral model in our experiments on the Discrim-Eval dataset (see Appendix C.2 and D.2.2 for details), we observe that our prompting strategies are more effective on the instruction-finetuned version compared to its base version. The intuition is that the effectiveness of prompting strategies in regulating biased pathways is strongly tied to the model's ability to follow instructions. We have updated our manuscript to include these discussions (Appendix C).

---

> > ### Author Response · Authors · 2024-12-01
> >
> > Dear reviewer qVWj,
> >
> > As the discussion period ends tomorrow, we believe we have addressed all the questions including the potential trade-off between bias reduction and reasoning capability in your initial review. Could you please clarify that? We are looking forward to your comments and feedback on our rebuttal. Thank you so much!

---

> > > ### Author Response · Authors · 2024-12-02
> > > **Discussion phase ending AOE today**
> > >
> > > Dear Reviewer,
> > >
> > > As the discussion phase quickly approaches an end, we are looking forward to your comments and feedback on our rebuttal. Thank you again for the time and effort!

---

### Official Review · Reviewer_qLJv · 2024-11-03

**Soundness:** 2
**Presentation:** 2
**Contribution:** 1
**Rating:** 3
**Confidence:** 3

**Summary:**

This paper addresses responsive bias in both open-source and proprietary LLMs by formulating a theoretical debiasing framework that analyzes the impact of social information on an LLM's decisions from a novel causal perspective. The framework identifies causal pathways through which social information influences model outputs and develops the integrated inference-time prompting strategies to accordingly regulate information flow across different pathways, thereby suppressing potential social bias. Extensive experiments on real-world datasets across multiple domains validate the framework, demonstrating its effectiveness in debiasing LLM decisions.

**Strengths:**

1. This paper provides a detailed analysis of the internal mechanisms behind biased decision-making in LLMs from a causal perspective, offering a theoretical understanding for the proposed prompting-based debiasing methods.
2. The Introduction section is well-written and effectively conveys the existing challenges.
3. Extensive experiments demonstrate that the proposed DDP significantly outperforms other baseline methods.

**Weaknesses:**

1. The proposed DDP suffers from poor applicability for general-purpose generation: DDP is a prompting-based technique that relies on adding extra textual guidance into  the prompt context to achieve debiasing. For evaluation, DDP needs dataset-specific design to obtain its corresponding textual guidance, thereby adapting to different evaluation datasets. So DDP can not be applied to free-form generation, especially when social attributes of interest are not directly given in input prompt but emerge as the intermediate generated results of LLMs, limiting its further applicability in more critical scenarios.
2. The paper organization is unclear and redundant: the pre-trained data generating process is not necessary for understanding the main contributions of this paper. The theoretical insights (strategies) are coupled together with the technical details, making it hard to fully and clearly assess the technical contributions after reading Section 3.
3. Prompting-based debiasing approaches have limitations and may not be particularly meaningful. Traditionally, the responsibility for model alignment and debiasing lies with model deployers. Implementing prompting-based debiasing can be unsafe and inevitably introduces additional time costs.
4. The textual guidance introduced by DDP may introduce noise that hinders the reasoning abilities of LLMs. As shown in Table 1-Type II, while DDP reduces the bias gap, it significantly degrades performance in terms of Anti and Pro (on GPT-3/3.5 and Claude 2). Therefore, their robustness remains an unsolved problem.

**Questions:**

1. Does the 'default' in Table 2 refer to the same meaning as the 'default' in Table 1, which solely receives the original question as input? If so, why does this method also include the metrics (TT, TF, FT, FF)?
2. See Weaknesses.

---

> ### Author Response · Authors · 2024-11-30
> **Reply Part I to Reviewer qLJv**
>
> We sincerely thank Reviewer qLJv for taking the time to review our work. While we appreciate the effort in providing feedback, we believe certain assessments do not fully reflect the contributions and intent of our study. We have respectfully addressed each point below to clarify misunderstandings:
>
> ---
>
> ### **C1:** "For evaluation, DDP needs dataset-specific design to obtain its corresponding textual guidance, thereby adapting to different evaluation datasets. So DDP can not be applied to free-form generation, especially when social attributes of interest are not directly given in input prompt but emerge as the intermediate generated results of LLMs, limiting its further applicability in more critical scenarios."
>
> **A1:** We **respectfully disagree** with the reviewer’s assessment. As explicitly stated in our abstract, introduction, and problem statement, the primary focus of this work is on mitigating bias in decision-making contexts, where outcomes have measurable, high-stakes implications. Addressing free-form generation is outside the scope of this study. Critiquing our work based on this perceived limitation overlooks the significant contributions our proposed framework and method offer in ensuring fairness in high-stakes decision-making contexts.
>
> Many leading works on bias mitigation, including established benchmarks like WinoBias, BBQ, and Discrim-Eval, similarly concentrate on decision-making rather than unconstrained text generation. This alignment underscores the relevance and impact of our contributions. Moreover, the decision-making framework evaluated in our experiments spans a broad range of critical applications, including hiring, healthcare, and education, which rely heavily on unbiased outcomes to ensure fairness and compliance with societal and legal norms. This makes our framework relevant to real-world use cases.
>
> The claim that "social attributes of interest are unavailable in practical settings" is also not entirely accurate. In numerous decision-making contexts, these attributes are often explicitly defined and governed by legal or corporate policies, such as:
> - **Legal frameworks** often mandate explicit definitions of protected attributes like race, gender, and age to ensure compliance with anti-discrimination laws (e.g., in employment or housing).
> - **Corporate policies** frequently require defining these attributes to support fairness and accountability in automated decision-making systems (e.g., adherence to equal opportunity statements).
>
> Given this context, our focus on regulated decision scenarios is not a limitation but a pragmatic and impactful choice tailored to real-world applications. While the adaptation of our framework to free-form generation remains an exciting avenue for future research, the scope of this work is intentionally focused on addressing bias in decision-making—an area of pressing societal importance.
>
> ---
>
> ### **C2:** "The pre-trained data generating process is not necessary for understanding the main contributions of this paper"
>
> **A2:** Thanks for the comment. We **respectfully disagree** with the reviewer on this point, and please allow us to clarify why the modeling of the pre-trained data-generating process is necessary.
>
> We base our work on the assumption that a well-trained and well-aligned LLM captures the dependence patterns in its training data and that such patterns are internalized and utilized during reasoning. This assumption is mild but fundamental---without it, not only would the LLM not function properly, but also the need for debiasing itself would be questionable (Reviewer `LwQ1` also kindly supported this).
>
> While this connection may seem intuitive, we believe it is important to explicitly include it to ensure the completeness of our work. This ensures that our contributions are grounded in the real-world data-generating process. By doing so, we also provide a solid foundation to link the observed training biases to the causal mechanisms that underpin our debiasing strategies.
>
> ---
>
> ### **C3:** "The theoretical insights (strategies) are coupled together with the technical details, making it hard to fully and clearly assess the technical contributions after reading Section 3."
>
> **A3:** Thanks for sharing the thought. In Section 3, we indeed organize the material in a strictly unified scheme:  presenting the theoretical underpinnings, introducing the corresponding debiasing strategies, and providing concrete example prompts for each strategy. This structure is intentional, as our debiasing strategies are directly motivated by and closely connected to the theoretical insights. Each strategy reflects specific aspects of the causal pathways we aim to regulate, and separating theory from strategy might weaken these connections.
>
> Please kindly let us know if there is still content that you would recommend organizing in a different way. Our goal is to ensure that the technical contributions are clear and fully assessable.
>
> ---

---

> ### Author Response · Authors · 2024-11-30
> **Reply Part II to Reviewer qLJv**
>
> ### **C4:** "Prompting-based debiasing approaches have limitations and may not be particularly meaningful. Traditionally, the responsibility for model alignment and debiasing lies with model deployers. Implementing prompting-based debiasing can be unsafe and inevitably introduces additional time costs."
>
> **A4:** We **respectfully disagree** with the reviewer’s characterization of prompting-based debiasing as limited or unsafe. On the contrary, extensive research supports the efficacy and meaningfulness of prompting techniques in addressing biases in large language models (LLMs). Prominent studies, such as Si et al. (2022), Tamkin et al. (2023), and Ganguli et al. (2023), have demonstrated that prompting-based methods can effectively reduce biases by leveraging the inherent knowledge encoded in LLMs without altering model parameters. These approaches are particularly valuable for mitigating bias in closed-source, black-box models like GPT-4, where fine-tuning is infeasible due to access constraints.
>
> We also advocate for a shared responsibility in bias mitigation. While model deployers undoubtedly have a crucial role in ensuring ethical usage, relying solely on them may overlook conflicts of interest (e.g., prioritizing business goals over fairness). By equipping end-users or intermediary systems with cost-effective debiasing tools, such as our proposed framework, we enable broader participation in bias mitigation. Fine-tuning, although effective, is often prohibitively expensive, particularly for smaller organizations or individual users. Prompting-based debiasing methods are indeed the scalable and adaptable alternative.
>
> Also, we should not be solely dependent on the model deployer to debias the models (model developers may have other interests, i.e., business, over bias mitigation); every party who is using or plans to use LLMs for consequential decision-making should take extra steps to help achieve fairer decisions. Finetuning these LLMs may be very costly and not affordable by individual parties, prompting-based debiasing is actually the cost-effective approach.
>
> Furthermore, our proposed Dual Directional Prompting (DDP) method can also be used to identify pairs of positive (unbiased) and negative (biased) responses by leveraging the intuition that an unbiased response should align with the model’s base decision. These contrastive pairs can be used to train reward models, aligning model outputs with fairness objectives. We have updated our manuscript to elaborate on these possibilities for future work.
>
> ---
>
> ### **C5:** "Does the 'default' in Table 2 refer to the same meaning as the 'default' in Table 1, which solely receives the original question as input? If so, why does this method also include the metrics (TT, TF, FT, FF)?"
>
> **A5:**  Thank you for the question! To clarify, the "Default" in Table 2 refers to the same method as the "Default" in Table 1, where the model is provided only the original question as input, without any additional debiasing prompts. The inclusion of the TT, TF, FT, and FF metrics in Table 2 is intended to offer a finer-grained error analysis of the model's performance. As detailed in lines 465–473, these metrics help us attribute errors to specific causes: whether they stem from the model's gender biases (TF) or limitations in non-gender-related world knowledge (FF).
>
> To compute the metrics in Table 2, we first ask the model the base question to evaluate its understanding or reasoning on the neutral scenario. Then, we apply the four methods (DDP, Fact Only, Counteract Only, and Default) to obtain the model's answer to the original question, and categorize the answers to the original question into the above four categories (TT, TF, FT, FF).
>
> We hope this explanation clarifies your questions.
>
> ---

---

> ### Author Response · Authors · 2024-11-30
> **Reply Part III to Reviewer qLJv**
>
> ### **C6:** "The textual guidance introduced by DDP may introduce noise that hinders the reasoning abilities of LLMs. As shown in Table 1-Type II, while DDP reduces the bias gap, it significantly degrades performance in terms of Anti and Pro (on GPT-3/3.5 and Claude 2). Therefore, their robustness remains an unsolved problem."
>
> **A6:** This is a great observation. The performance drop observed in Table 1 indeed does not represent a trade-off between bias reduction and reasoning capability but rather represents **a recalibration of the model's reliance on biased shortcuts**.  As supported by our detailed ablation studies in Table 2, this reduction does not stem from hindered reasoning capabilities but from a deliberate intervention to reduce dependency on using social stereotypes.
>
> For instance, in Table 2, when GPT-3.5 was asked base questions—neutral reformulations designed to remove gendered pronouns and test world-knowledge-based reasoning—it answered 19.13% incorrectly  (19.13% is the sum of errors from "FT-Pro" and "FF-Pro" or equivalently "FT-Anti" and "FF-Anti" categories, e.g., 5.97% + 13.15%). However, However, when the original pro-stereotype questions associated with these same base questions were asked directly (Default), GPT-3.5 answered approximately 83% (15.88%/19.13%) of them correctly.  This high accuracy on pro-stereotype questions, despite errors in the neutral base questions, reveals the model’s strong reliance on biased gender shortcuts.
>
>
> By applying DDP, we reduced this reliance from 15.88% to 5.97%. This reduction aligns with a decrease in GPT-3.5's performance on pro-stereotype questions from 94.03% to 84.67% (Table 1). Importantly, this outcome demonstrates that DDP effectively mitigates biased reasoning pathways without compromising the model's intrinsic reasoning capabilities.
>
> The base question plays a pivotal role in this analysis by decoupling the model’s performance from social stereotypes and isolating its reasoning ability. By highlighting the model’s divergence between base and original questions, we underscore how DDP effectively regulates the model's reasoning pathways to nudge it away from shortcut-based (biased) reasoning.

---

> > ### Author Response · Authors · 2024-12-01
> >
> > Dear reviewer qLJv,
> >
> > As the discussion period ends tomorrow, we believe we have addressed all the questions and concerns in your initial review. Could you please clarify that? We do hope you can take your time to read through our detailed reply, and we are more than happy to have further discussions if needed. Thank you so much for your time and help!

---

> ### Author Response · Authors · 2024-12-02
> **Discussion phase ending AOE today**
>
> As the discussion phase quickly approaches an end, we are eager to understand if our **point-by-point responses** and **extensive additional experiments as per request** help address the questions and concerns, especially potential misunderstandings. Thank you again for your service.

---

### Official Review · Reviewer_QvwJ · 2024-11-05

**Soundness:** 2
**Presentation:** 3
**Contribution:** 2
**Rating:** 6
**Confidence:** 5

**Summary:**

This paper introduces a prompt-based method to remove biases in a language model's output. It motivates the prompts using the idea of selection bias from causal inference literature. Experiments show a significant reduction in measured bias on two datasets.

**Strengths:**

* Theoretical justification of prompting strategies using a stylized selection bias argument
* Significant reduction of bias in demonstrated experiments.

**Weaknesses:**

* The connection of selection bias, causal graph and the proposed prompt is weak. In one dataset, strategy I and II is used. In the other dataset, strategy I and III. So it is not clear that the set of selection biases listed are exhaustive. Rather, it appears that the list of such selection biases may increase as the datasets increase, and so the current selection biases presented are the union of those that were useful for prompts for these two data sets.
* Ablations are not provided. What happens when all strategies are used for first dataset, and what happens when strategy I and II is used for second dataset?
* Other methods of removing bias are not compared. Since the final solution is just a prompting change, there can be other ways of prompting the LLM that are simple to implement. For example, the text "avoid any gender bias while answering the question" can be added to the prompt.
* Strategy I seems the most useful. However, I have two concerns. First, it may be computationally intensive. Can the authors clarify how many calls would it need to answer a single question? If I understand correctly, there will be one call to create the two possible scenarios for the base question, and then there will be one call to decide which of them is more plausible, and then a final call to actually answer the question? So there are going to be three calls per question? If that's the case, I can think of other methods, for example, the React multi-agent framework, where the LLM is asked to respond to the question first, then there is a critique agent that checks whether there is any bias in the answer, and if yes, it asks the LLM to regenerate the answer giving the feedback from the critique as a part of the prompt. Second concern is that the creation of the base question, while could be templated for simple examples like the one shown in the paper, but eventually will also become a task that an LLM will need to do so, are there ways to use a smaller model or something more efficient for this?

**Questions:**

I have mixed opinions about this paper. On the one hand, I appreciate the selection bias analogy and the abstraction of the problem to a causal graph and the conclusions that come from it. On the other hand, the final solution proposed is just an ensemble of intuitive prompts, and it is not clear to me that these prompts are meaningfully guided by the selection bias theory. And it is not clear to me that other better prompts could not have been obtained without any selection bias theorization. So my questions to the authors are :
1. Can you show that avoiding bias based on selection bias is better than simply asking the LLM to avoid bias? See one suggestion of a modified prompt in the weaknesses above. I guess the literature on debiasing LLMs may have more simple prompt additions or system prompts that can be added.  In your comparison, also consider the computational cost of your proposed method.
2. How does the selection bias theory help in choosing the correct strategy, and later, the prompt? What was the justification for using strategy II in one case and strategy III in another?
3. How would Strategy I generalize to more complex bias scenarios? Do you need an LLM for creating the base prompt in that case? Many cases of biases may not be simple template pronoun substitutions.
4. Strategy I seems closest to a variant of in-context-learning where the closest example is chosen so to help the LLM answer the question correctly. Can you compare to a baseline that selects the K nearest in context examples to add to the prompt, rather than a fixed set of in context examples?

---

> ### Author Response · Authors · 2024-11-30
> **Reply Part I to Reviewer QvwJ**
>
> Thanks for the detailed and thoughtful comments and questions, as well as the time and effort devoted! There might be some misunderstandings, and below please allow us to provide point-by-point responses:
>
>
> ---
>
> ### **C1:** "It is not clear that the set of selection biases listed are exhaustive. Rather, it appears that the list of such selection biases may increase as the datasets increase, and so the current selection biases presented are the union of those that were useful for prompts for these two data sets."
>
> **A1:** We acknowledge the reviewer's observation regarding the exhaustiveness of the selection biases addressed in our work. Our intent is not to claim that our example prompts represent an exhaustive set covering all selection biases. Rather, we provide an adaptable framework: we view biases as a reflection of societal interests, norms, and assumptions, which can evolve over time. For instance, our focus on mitigating biases involving social categories (e.g., gender) is motivated by prevailing concerns in current socio-cultural contexts where such biases are widely recognized as objectionable.
>
> When societal values shift and new awareness emerges—potentially highlighting biases in areas previously overlooked (e.g., cognitive attributes like IQ or other individual traits)—our framework remains adaptable. Our methodology generalizes to mitigate biases as they emerge, by analyzing the causal pathways through which they operate. This ensures that as societal values evolve, our framework can seamlessly address new biases without requiring structural changes, demonstrating its scalability and robustness in promoting fairness across diverse contexts.
>
> ---
>
> ### **C2:** "Ablations are not provided. What happens when all strategies are used for first dataset, and what happens when strategy I and II is used for second dataset?"
>
> **A2:** As strategies II and III both aim at discouraging biased reasoning, a focus of prior literature, our experiments have mainly focused on exploring how the novel strategy I (i.e., encouraging fact-based reasoning) complements existing prompting-based bias mitigation techniques. This is why we name our method Dual-Directional Prompting (DDP): it combines the prompting strateg(ies) to discourage biased reasoning with the strategy to encourage bias-free reasoning.
>
> Still, to address your query and out of curiosity, we have conducted additional experiments incorporating all three strategies, as well as ablation studies on the Winobias dataset. These results are now included in Table 6 of the updated manuscript. As we expected, simply telling the model not to use gender-related information (Strategy III) doesn't help much with mitigating the bias, combining all three strategies achieves slightly better performance.
>
> In addition, regarding your interest in combining Strategy I and II, we would like to direct your attention to the ablation study presented in Table 7 of the updated manuscript, where we systematically adjusted the extent to which we counteract existing selection bias. Please refer to the detailed analysis provided in lines 1062–1073 of the updated manuscript.
>
> We sincerely appreciate your thoughtful suggestion, which has helped us strengthen the robustness of our conclusions with these additional results.
>
>
> ---
>
> ### **C3:** "It is not clear to me that these prompts are meaningfully guided by the selection bias theory"
>
> **A3:** Thank you for engaging deeply with our work and considering how prompts are guided by the selection bias theory. When introducing our prompting strategies in Section 3.3, for each strategy, we present both the theoretical foundation—explaining the objectives and the selection mechanism(s) underpinning each strategy (e.g., lines 246--249, 261--264, 279-281)—and the corresponding example prompts (e.g., lines 256--259, 274--275, 288--289) in blue-colored text. The prompt designs are directly guided by these strategies, ensuring they are closely aligned with the selection mechanisms employed in each case.
>
> ---

---

> ### Author Response · Authors · 2024-11-30
> **Reply Part II to Reviewer QvwJ**
>
> ### **C4:** "Other methods of removing bias are not compared. Since the final solution is just a prompting change, there can be other ways of prompting the LLM that are simple to implement. For example, the text "avoid any gender bias while answering the question" can be added to the prompt."
>
> **A4:** Thank you for the insightful suggestion regarding alternative prompt strategies to mitigate bias. In fact, as part of our work, we conducted comprehensive experiments on the Discrim-Eval dataset introduced by Anthropic [1] (details in Appendix C.2), where we systematically evaluated multiple instantiations of the same prompting strategy.
>
> Our findings, presented in Appendix D.2.2 due to space limit, demonstrate that while simplistic prompts like the one suggested can reduce bias to some extent, no single instantiation (i.e., prompt example) stands out as the universally most effective one across various LLMs and demographic categories. Instead, combining the strategies of encouraging fact-based reasoning with discouraging biased reasoning (DDP) yields a more significant reduction in the relative bias gap compared to applying either approach in isolation.
>
> [1] Tamkin, Alex, et al. "Evaluating and mitigating discrimination in language model decisions." arXiv preprint arXiv:2312.03689 (2023).
>
> ---
>
> ### **C5:** "Strategy I seems the most useful. However, I have two concerns. First, it may be computationally intensive. Can the authors clarify how many calls would it need to answer a single question? ... Second concern is that the creation of the base question, while could be templated for simple examples like the one shown in the paper, but eventually will also become a task that an LLM will need to do so, are there ways to use a smaller model or something more efficient for this?"
>
> **A5:** Thank you for highlighting these points. Below, we address the computational efficiency and general applicability of Strategy I:
>
> 1. Computational Efficiency on our current approach:
>
> In our experiments with datasets like WinoBias, BBQ, and Discrim-Eval, the creation of the base question is automated using regular expressions, resulting in only **one additional call** per individual question to obtain the model’s answer to the base question.
>
> For general applications, it is possible that we may need one additional call to generate the base question. However, the same base question is often shared across multiple individual questions. For instance, in datasets like BBQ and Discrim-Eval, the same decision scenario is tested across different genders, different ethnicities, and so on. In such cases, the additional cost of generating the base question and obtaining the answer to it is **effectively averaged across all related individual questions that share the same base question, making the computational overhead negligible.** This reuse mechanism also gives our method an advantage over multi-agent frameworks like ReAct. In ReAct, the additional calls occur after posing the original question and cannot be shared across different individual questions.
>
>
> 2. Exploring Efficiency Improvements:
>
> We agree that employing a smaller model to generate the base question could significantly enhance efficiency. This approach could involve fine-tuning a lightweight model specifically for this task or leveraging pre-trained smaller models to handle base question generation. Exploring such methods is a promising direction for future research. We have incorporated these discussions in our updated manuscript.
>
> ---
>
> ### **C6:** How does the selection bias theory help in choosing the correct strategy, and later, the prompt? What was the justification for using strategy II in one case and strategy III in another?
>
> **A6:** We appreciate the insightful question. The selection bias theory informs our choice of strategy by identifying which causal pathways need regulation. Below, we clarify our approach and rationale in two parts:
>
> - We are not claiming one strategy is better than another. Individual debiasing strategies are only effective to a certain extent, but combining them is often more effective (when conditions permit). We have presented the theoretical characterization and provided remarks to discuss this point in detail (Theorem 3.1).
>
> - For suitability, Strategy III is broadly applicable as long as we know what kind(s) of social information should not influence the LLM's decision. In contrast, Strategy II requires additional knowledge of specific entities or scenarios we are dealing with (e.g., career) as it directly counteracts the bias introduced by the selection mechanism linking social category information to entities or scenarios.
>
> ---

---

> > ### Author Response · Authors · 2024-11-30
> > **Reply Part III to Reviewer QvwJ**
> >
> > ### **C7:** How would Strategy I generalize to more complex bias scenarios? Do you need an LLM for creating the base prompt in that case? Many cases of biases may not be simple template pronoun substitutions.
> >
> > **A7:** We appreciate the reviewer’s concern and would like to emphasize that Strategy I has already been evaluated in complex bias scenarios, as demonstrated in our experiments on the Discrim-Eval dataset. Discrim-Eval comprises 70 diverse decision scenarios, which inherently reflect a broad spectrum of bias contexts. Many individual questions in this dataset share a common base scenario, allowing us to effectively utilize templates or regular expressions to extract the shared neutral component to create the base question. This approach highlights Strategy I's capability to generalize beyond simple template-based substitutions.
> >
> > ---
> >
> > ### **C8:** Strategy I seems closest to a variant of in-context-learning where the closest example is chosen so to help the LLM answer the question correctly. Can you compare to a baseline that selects the K nearest in context examples to add to the prompt, rather than a fixed set of in context examples?
> >
> > **A8:** This is an interesting hypothesis, and Table 3 of [2] provides relevant insights. In their analysis, various sets of ICL examples were constructed (e.g., examples exclusively from anti-bias scenarios for Type I questions). The findings reveal that while using ICL examples from a specific category enhances performance for that category (e.g., pro-bias examples improve pro-bias predictions), a balanced set of pro-bias and anti-bias examples proved most effective in reducing bias. Notably, this balanced strategy is one of the baselines we compare against. We have incorporated this discussion into the updated manuscript (Appendix C.1).
> >
> > [2] Si, Chenglei, et al. "Prompting GPT-3 to be reliable." https://arxiv.org/pdf/2210.09150 ICLR 2023

---

> ### Author Response · Authors · 2024-12-01
>
> Dear reviewer QvwJ,
>
> As the discussion period ends tomorrow, we believe we have addressed all the questions and requested additional ablation experiments in your initial review. Could you please clarify that? We are looking forward to your comments and feedback on our rebuttal. Thank you so much!

---

> > ### Author Response · Authors · 2024-12-02
> > **Discussion phase ending AOE today**
> >
> > Dear reviewer,
> >
> > As the discussion phase quickly approaches an end, we are eager to understand if our point-by-point responses and extensive additional experiments as per request help address the questions and concerns. We are looking forward to your comments and feedback on our rebuttal. Thank you again for the time and effort!

---

> > > ### Comment · Reviewer_QvwJ · 2024-12-03
> > > **thanks for the response**
> > >
> > > Thanks for the detailed response. Many of my queries are answered and I'm happy to raise my score.

---

> > > > ### Author Response · Authors · 2024-12-03
> > > > **Thank Reviewer QvwJ for the Feedback**
> > > >
> > > > Dear `Reviewer QvwJ`,
> > > >
> > > > Thanks for getting back to us, and for the encouraging acknowledgement.
> > > >
> > > > Please just feel free to let us know if you would like to suggest any further changes.
> > > >
> > > > Yours sincerely,
> > > >
> > > > Authors of `Submission 12960`

---

### Author Response · Authors · 2024-11-30

To all reviewers,

We sincerely appreciate your time and effort in reviewing our work! We have carefully addressed all your comments and provided detailed responses in the rebuttal phase. Additionally, we have updated our manuscript to reflect these improvements. You can access the updated manuscript through this [anonymous link](https://file.io/x4qcO5DjnpGe).

We look forward to any further insights you might have, and we would love to continue further discussions.

Best regards,

Authors of Submission 12960

---

### Meta-Review · Area_Chair_Yicp · 2024-12-18

**Metareview:**

The paper presents a novel framework that leverages causal analysis to mitigate biases in large language models (LLMs) through prompting strategies. The paper’s strengths include its theoretical foundation, extensive experimental results demonstrating bias reduction, and a comprehensive bias-mitigation solution.

However, the paper has some weaknesses, such as a weak connection between selection bias, causal graphs, and proposed prompts, limited discussion on the effectiveness of the approach with different model scales and architectures, and concerns about computational intensity and general applicability. The authors addressed these concerns by providing additional experiments, clarifying theoretical foundations, and emphasizing the scalability and adaptability of their approach.

The authors’ responses to the reviewers’ questions were thorough and addressed many of the raised concerns, leading to an improved understanding of the paper’s contributions. One reviewer raised some concerns about the applicability of the method. The authors provided reasonable attempts to respond to the questions; however, the reviewer was not responsive.

Given these considerations, I suggest accepting this paper,

**Additional Comments On Reviewer Discussion:**

The authors’ responses to the reviewers’ questions were thorough and addressed many of the raised concerns, leading to an improved understanding of the paper’s contributions. One reviewer raised some concerns about the applicability of the method. The authors provided reasonable attempts to respond to the questions; however, the reviewer was not responsive.

---

### Decision · Program_Chairs · 2025-01-22

Accept (Poster)